

# On the linkage between future Arctic sea ice retreat, Euro-Atlantic circulation regimes and temperature extremes over Europe

Johannes Riebold[1], Andy Richling[2], Uwe Ulbrich[2], Henning Rust[2], Tido Semmler[3], and Dörthe Handorf[1]

[1]Alfred-Wegener-Institut, Helmholtz-Zentrum für Polar- und Meeresforschung, Potsdam, Germany
[2]Institute for Meteorology, Freie Universität Berlin, Berlin, Germany
[3]Alfred-Wegener-Institut, Helmholtz-Zentrum für Polar- und Meeresforschung, Bremerhaven, Germany

**Correspondence:** Johannes Riebold (Johannes.riebold@awi.de)

**Abstract.** The question to what extent Arctic sea ice loss is able to affect atmospheric dynamics and climate extremes over mid-latitudes still remains a highly debated topic. In this study we assess the impact of future Arctic sea ice retreat on occurrence probabilities of wintertime circulation regimes and link these dynamical changes to frequency changes in European winter temperature extremes. For this reason, we analyze ECHAM6 sea ice sensitivity model simulations from the Polar Amplification

Intercomparison Project and compare experiments with future sea ice loss prescribed over the entire Arctic, as well as only locally over the Barent/Karasea with a present day reference experiment. We first show how these imposed future Arctic sea ice reductions affect large-scale atmospheric dynamics in terms of occurrence frequency changes of five computed Euro-Atlantic winter circulation regimes. Both sensitivity experiments show similar regime frequency changes, such as more frequent occurrences of a Scandinavian blocking pattern in midwinter under reduced sea ice conditions. Afterwards we demonstrate how

the Scandinavian blocking regime, but also a regime that resembles the negative phase of the North Atlantic Oscillation can be linked to favored occurrences of European winter cold extremes. In contrast, winter warm extreme occurrences are typically associated with an anticyclonic regime over the eastern Atlantic and a regime similar to the positive state of the North Atlantic Oscillation. Based on these links between temperature extremes and circulation regimes, as well as on the previously detected regime frequency changes we employ a framework of conditional extreme event attribution. This enables us to decompose

sea ice induced frequency changes of European temperature extremes into two different contributions: one term that is related to dynamical changes in regime occurrence frequencies, and another more thermodynamically motivated contribution that assumes fixed atmospheric dynamics in terms of circulation regimes. By employing this decomposition procedure we show how the overall thermodynamical warming effect, but also the previously detected increased Scandinavian blocking pattern frequency under future sea ice reductions can dominate and shape the overall response signal of European cold extremes in

midwinter. We also demonstrate how for instance a decreased occurrence frequency of the anticyclonic regime over the eastern Atlantic counteracts the thermodynamical warming response and results in no significant changes in overall January warm extreme occurrences. However, when compared to other characteristics of future climate change, such as the thermodynamical impact of globally increased sea surface temperatures, we argue that the detected effects on European temperature extremes related to Arctic sea ice loss are of secondary relevance.



## 1 Introduction

Recent global warming includes a phenomenon called Arctic Amplification that comes along with an up to four times faster warming of Arctic regions compared to global average over recent decades (Rapanen et al., 2021). This amplified Arctic warming is predominantly observed in winter time and is accompanied by an unprecedented shrinkage of Arctic sea ice concentration and thickness (Stroeve and Notz, 2018). Model projections forced under different greenhouse gas scenarios show clear evidence of a continuation of sea ice decline, with some models suggesting a seasonally ice-free Arctic till the mid of the century (Notz and Coauthors, 2020) . Aside from local ecological and economical impacts (Meredith et al., 2019) the question to what extent Arctic climate change and related sea ice loss may impact mid-latitude weather and general atmospheric dynamics has received a lot of attention over the last years and decades (e.g. Cohen et al., 2020; Screen, 2017b; Handorf et al., 2015; Cohen et al., 2014). A large variety of potential hemispheric-wide atmospheric responses have been detected and hypothesized in connection to Arctic sea ice loss. Such responses include for instance a commonly observed negative winter NAO response (e.g. Screen, 2017b; Nakamura et al., 2015; Jaiser et al., 2012), a highly debated weakening and stronger meandering of the jet stream that may result in more stationary and slower propagating large-scale Rossby waves (Francis and Vavrus, 2012; Barnes and Screen, 2015; Riboldi et al., 2020), as well as an intensification of the Scandinavian and Ural highs leading to continental winter cooling over Eurasia (Cohen et al., 2018). In this respect, dynamical pathways have been proposed relating for instance sea ice and snow cover anomalies in autumn to enhanced vertical wave activity fluxes and a weakened stratospheric polar vortex. These stratospheric disturbances could subsequently propagate downward and finally result in a late winter negative NAO response (Cohen et al., 2014; Nakamura et al., 2016; Jaiser et al., 2016; Sun et al., 2015). Especially the Barent/Karasea region, being a hotspot of recent Arctic sea ice retreat, has been argued to play an essential role for triggering such dynamical pathways (Screen, 2017a; Jaiser et al., 2016; Kretschmer et al., 2016). Nevertheless, no overall consensus about linkages and the underlying dynamical pathways has been reached until now (Cohen et al., 2020), mostly due to discrepancies between observational and modeling studies. A recent study by Siew et al. (2020) highlighted for instance that the intermittent and state-dependent character of the aforementioned stratospheric pathway might be a potential reason for the typical low signal-to-noise ratios of atmospheric responses to sea ice changes. Furthermore, Petoukhov and Semenov (2010) showed how the modeled atmospheric response can depend on the magnitude of prescribed sea ice loss in the Barent/Karasea in a highly nonlinear way. Although most studies on Arctic-midlatitude linkages focus on the role of sea ice changes, several recent studies (He et al., 2020; Labe et al., 2020) also highlighted the importance of the vertical extent of Arctic warming into the upper troposphere compared to sea ice loss alone.

From a more large-scale and regime-oriented perspective, atmospheric dynamics can be viewed in a variety of conceptual frameworks (Hoskins and Woollings, 2015) including for instance jet stream states, blockings or atmospheric circulation regimes. Especially the framework of circulation regimes has been employed in a large variety of studies (e.g. Crasemann et al., 2017; Horton et al., 2015) in order to characterize the atmospheric circulation. Circulation regimes provide physically meaningful categorizations (Hochman et al., 2021) of atmospheric low-frequency variability into different regime states and have also been considered as preferred or quasi-stationary states of the underlying nonlinear atmospheric system (Hannachi





et al., 2017). It has been hypothesized that weak external forcings imposed to the atmospheric system are able to modify the

occurrence probability of such regime states (Corti et al., 1999), while not effecting the overall regime structure (Palmer, 1999). Indeed, Crasemann et al. (2017) compared atmosphere-only model experiments forced under low and high sea ice conditions relative to the recent past and showed how the occurrence probability of certain Euro-Atlantic circulation regimes can be significantly affected by such Arctic sea ice changes. In this case the induced sea ice changes were considered as weak forcings applied to the atmospheric system.

A major interest for human society nowadays is given by the question to what extent the recently observed increasing number of climate extremes (Coumou and Rahmstorf, 2012) can be attributed and effected by anthropogenic global warming (Otto, 2016). Basically, there is an overall agreement that from a thermodynamical perspective global warming will lead to less (more) frequent and intense cold (warm) extremes. Nevertheless, the occurrence of cold spells like over Europe in 2010 (Cattiaux et al., 2010) or the more recent cold air outbreak over North America in 2021 (Bolinger et al., 2022) might be

considered as contradictions to this simplified thermodynamical perspective. In this respect, Cattiaux et al. (2010) illustrated for instance how the European winter cold spell in 2010 was, from a thermodynamical point of view, perfectly in line with recent global warming when accounting for the anomalous negative NAO situation during this winter. Shepherd (2016) framed a storyline approach aiming to unfold specific classes of extreme events into the different contributing factors by including both, dynamical and non-dynamical contributions. However, circulation changes found in climate model simulations typically

suffer low signal-to-noise ratios. Therefore, changes regarding the dynamical situation leading to a certain extreme should only be included into an analysis when there is solid evidence that changes in atmospheric circulation can be expected or reliably detected (Trenberth et al., 2015; Shepherd, 2016).

Since, as mentioned above, Arctic sea ice retreat has been proven to be potentially able to modify atmospheric large-scale dynamics, the question appeared how changes in mid-latitude weather can be dynamically and thermodynamically attributed

to Arctic sea ice changes. Screen (2017b) compared large ensembles of atmosphere-only experiments forced under low and high sea ice conditions relative to the recent past. They observed that despite an intensification of negative winter NAO events under low Arctic sea ice conditions an expected dynamically induced European cooling response was absent, mostly due to compensation effects related to an overall thermodynamical warming. Another study by Deser et al. (2016a) investigated how different complexities of an ocean model can affect the large-scale hemispheric circulation response to Arctic sea ice loss. They

compared model simulations with Arctic sea ice conditions constrained to the late 21th and to the 20th century. On the one hand they argued that under reduced sea ice conditions elevated sea level pressures over northern Siberia and Arctic regions are associated with anomalous northeasterly advection of cold Arctic air masses towards central Eurasia. This dynamically induces a cooling response over the respective central Eurasian regions. On the other hand, this dynamical cooling effect may be thermodynamically counteracted by elevated SSTs, which was especially the case for coupled ocean–atmosphere

model setups. Recent studies however argue that such coupled model setups artificially overestimate the impact of sea ice loss (England et al., 2022). Recently, Chripko et al. (2021) studied fully coupled model experiments where the sea ice albedo parameter was reduced to ocean value yielding mostly ice-free conditions from July to October and moderate sea ice reductions in winter. When compared to a control simulation they detected winter warming signals over Europe and North America in



the sensitivity experiment. By applying a dynamical adjustment method (Deser et al., 2016b) they showed that these overall
responses could be explained by a combination of a dynamical response and a residual contribution.

Based on such previous studies that decomposed and linked changes in mid-latitude weather and dynamics to Arctic sea
ice loss, as well as due to the high societal relevance of extreme events nowadays the question arises to what extent future
sea ice retreat is able to impact the occurrence of extreme weather events. Therefore, in this study we investigate the impact
of future Arctic sea ice loss on the mid-latitude circulation over the Euro-Atlantic domain and related European temperature
extremes. Here, we will focus on winter temperature extremes over the European region that can have significant impacts on
society (Díaz et al., 2005) and economy (Savić et al., 2014; Añel et al., 2017) over such densely populated regions. In order to
assess and isolate the impact of Arctic sea ice changes we will investigate ECHAM6 model runs from the Polar Amplification
Intercomparison Project (PAMIP). The experiments that are considered here are forced under present day and reduced future
sea ice conditions over the entire Arctic, as well as under sea ice conditions only locally reduced over the Barent/Karasea. The
latter allows for assessing the role of sea ice loss specifically in the Barent/Karasea region. Based on the detected circulation
changes, as well as by employing a framework of conditional extreme event attribution (Yiou et al., 2017) we will determine
circulation and non-circulation related contributions to overall changes in extreme occurrences. More specifically, the analysis
steps can be divided into different research questions and analysis steps that are partially linked to each other:

1. Within the methodological framework of atmospheric circulation regimes, what changes in the wintertime atmospheric
large-scale circulation over the Euro-Atlantic sector can be expected under future Arctic sea ice retreat?

2. Which regimes can be associated with preferred occurrences of winter temperature extremes over Europe?

3. What overall frequency changes of extreme occurrences over the continental Northern Hemisphere can be detected in
   response to future sea ice changes?

4. Based on the sea ice induced changes in circulation regimes, to what extent can frequency changes of European extremes
be related to circulation and non-circulation related contributions?

When studying the impact of Arctic sea ice changes on mid-latitude circulation and weather, the question may arise how such
impacts compare to atmospheric responses induced by other facets of future climate change. Therefore, in order to assess the
relative importance of sea ice loss on future changes in European extremes, the analysis will be complemented by investigating
the impact of a globally increased future SST background state prescribed in one of the experimental setups.

**2  Data**

In this study we analyze different ECHAM6 (high resolution setup with T127 and 95 vertical layers up to 0.01 hPa) sea
ice sensitivity simulation data from the Polar Amplification Intercomparison Project (PAMIP, Smith et al., 2019). The PAMIP
protocol aims on a better understanding of the impact and relative roles of Arctic sea ice and SST changes on the global climate
system. Therefore, each sensitivity experiment includes 100 ensemble members of one-year-long atmosphere-only time slice





simulations that are forced under different annual cycles of sea ice, but also SST boundary conditions. Initial conditions of each ensemble member are based on AMIP simulations for 1st April 2000 and each ensemble member was run for 14 months, but the first two months were finally excluded for model spin up reasons. In order to study the impact of future sea ice changes on circulation regimes and related changes in extremes we analyze sensitivity simulations forced under:

– present day SST and present day sea ice conditions (pdSI/pdSST, PAMIP setup 1.1)

– present day SST and future/reduced Arctic-wide sea ice conditions (futArcSI, PAMIP setup 1.6)

– present day SST and future/reduced sea ice in the Barent/Karasea region 65-85°N, 10-110°E (futBKSI, PAMIP setup 3.2).

In order to contrast the importance of future SST with Arctic sea ice changes in the very end of this study, we also consider a sensitivity simulation forced under

– present day sea ice and globally raised future SST conditions (futSST, PAMIP setup 1.4).

The pdSI/pdSST simulation serves in a first place as a reference simulation to which the sensitivity simulations futArcSI and futBKSI are compared with. Comparisons of sea ice and SST forcing fields of the respective present day and sea ice sensitivity simulations are shown in Smith et al. (2019) Figs. 5 and 6. In winter, future sea ice conditions are predominantly reduced over the Barent/Karasea, the Sea of Okhotsk, the Bering Sea and parts west and east of Greenland. Summer conditions are 140 characterized by strong reductions and ice-free areas over central Arctic regions.

Present day forcing fields are obtained from the climatologies of observations from the Hadley Centre sea ice and Sea Surface Temperature dataset over the period 1979–2008 (Rayner et al., 2003). Future conditions are derived from RCP8.5 multimodel simulations for a 1.43 (2)°C warming scenario over present day (preindustrial) conditions (for more details see Smith et al. (2019) Appendix A). At grid points where sea ice has been removed under future conditions the present day SSTs are replaced 145 by future SSTs if the difference in sea ice concentration between future and present day is greater than 10 %. Sea ice thickness at each grid point is set to 2 m for all simulations and greenhouse gas forcings are constantly set to present day conditions of the year 2000.

For the analysis presented in this study we use daily sea level pressure (slp), as well as daily maximum/minimum near-surface air temperature (tasmax/tasmin). The daily temperature and slp data are provided on a regular lon-lat grid with 0.9375° 150 resolution, however, the slp data are finally regridded to a 100×100 km equal-area grid (see also next Section). In order to complement and backup certain parts of our analysis with real world data, we additionally used slp and 2 meter temperature data from the ERA5 reanalyses over the period 1979–2018 (Hersbach et al., 2020).



## 3  Methods

### 3.1  Circulation regimes

In this study, we compute centroids $C_i$ of atmospheric circulation regimes for the extended winter season with the $k$-means clustering algorithm (Michelangeli et al., 1995; Crasemann et al., 2017; Straus et al., 2007) applied to daily slp anomaly data merged together from two different experiments (typically the pdSI/pdSST reference simulation and one of the sensitivity simulations) over the Euro-Atlantic domain (90°W-90°E,20°N-88°N). Before applying the clustering algorithm, slp data were regridded to a $100 \times 100$ km equal-area grid in order to avoid grid point convergence at higher latitudes. Generally speaking

$k$-means clustering aims to minimize the intra-cluster to inter-cluster variance ratio by an iterative allocation and exchange procedure of cluster members (MacQueen, 1967). In order to reduce computational demands, we applied a dimensionality reduction via PCA prior to the clustering algorithm. Here we used the first 20 Principal components that roughly explain around 90% of variance of the winter slp anomaly fields. Further increasing the number of PCs did not effect the final outcome of the clustering algorithm. The $k$-means algorithm has been initialized for 1000 times and the best result in terms of minimizing the

aforementioned variance ratio has been finally chosen. Based on the Euclidean distance, the respective slp anomaly field or atmospheric flow $F$ at each day is finally assigned to the best-matching cluster centroid $C_i$.

Slp anomalies are generally calculated as deviations from an annual cycle, which is obtained by averaging each day of a year over all years. For the merged pdSI/pdSST+futBKSI and pdSI/pdSST+futArcSI datasets we computed a joint annual cycle of both simulations. It shows that the resulting cluster allocations are not considerably affected by whether the slp anomalies have been calculated as deviations from the joint annual cycle as described above, or by removing the annual cycles for

each experiment individually (as done by e.g. Crasemann et al., 2017). This is also related to the fact that when contrasting the reference with both sea ice sensitivity experiments the respective winter slp background states showed mostly negligible differences, neither did they project on any mode of variability. In contrast to the sea ice sensitivity simulations, the relatively strong forcing in the futSST experiment leads to an evident change of the slp background state (with respect to the reference

simulation) that strongly projects on a negative NAO pattern. This background difference pattern significantly affects the final cluster allocations when subtracting a joint annual cycle. Therefore, we computed the annual cycle for both simulations individually when merging data from the futSST and the pdSI/pdSST experiments to take into account the different background states.

A subtle part when applying cluster algorithms as $k$-means is to prescribe the number of clusters and therefore make an

assumption about the number of existing atmospheric circulation regimes beforehand. Several attempts have been made in order to determine such an optimal number of winter regimes with most studies indicating a number between four and six regimes (Falkena et al., 2020). Here we stick to a cluster number of five which is supported by recent studies (Crasemann et al., 2017; Dorrington and Strommen, 2020).



### 3.2 Conditional extreme event attribution framework

In this study we also aim to identify thermodynamical and dynamical contributions to overall European temperature extreme frequency changes in the sea ice sensitivity experiments. Dynamically induced changes in the occurrence frequencies of certain local extreme events are related to changes in the relevant dynamical conditions, e.g. in terms of more frequent occurrences of the respective atmospheric flow patterns that promote a certain extreme. In contrast, thermodynamical contributions are typically associated with changes of extreme probabilities that would also occur in the absence of any relevant dynamical

changes (e.g. due to overall global warming). From a methodological point of view it is however challenging to clearly separate dynamical and thermodynamical components. This issue is related to the fact that there is generally no unique way to define and detect changes in all contributing dynamical and non-dynamical factors that impact a certain class of extreme event.

   Nevertheless, a variety of approaches have been outlined over the years (e.g. Yiou et al., 2017; Deser et al., 2016b; Vautard et al., 2016; Cassano et al., 2007) that aim to decompose atmospheric responses into thermodynamical and dynamical contribu-

tions. In this study a framework for conditional extreme event attribution (Yiou et al., 2017) is utilized. This method provides a suitable approach for decomposing changes in extreme event occurrence frequencies while employing the framework of circulation regimes.

   In this study winter extreme events are defined as exceedances (or falls below) of a threshold temperature $T_{\mathrm{ref}}$ that is based on the 100 simulated winters in the reference pdSST/pdSI simulation. The threshold temperature $T_{\mathrm{ref}}$ of warm (cold) extreme

events at a given grid point is computed for each winter month separately as the 0.95 (0.05) quantile of the respective underlying daily tasmax (tasmin) distribution in pdSST/pdSI.

   Based on this definition we define the probabilities $p_0$ and $p_1$ in a counterfactual and factual world, respectively, of a warm (cold) extreme occurrence at a certain grid point as

$$p_{0/1} = \mathrm{Pr}(T_{0/1} \lessgtr T_{\mathrm{ref}}) \tag{1}$$

where $T_0$ is the temperature in the counterfactual world and $T_1$ in the factual world. In this study, we define the factual world (the world as it is) as the pdSST/pdSI reference simulation. The counterfactual world (a world that might occur) is given by the different ECHAM6 PAMIP sea ice sensitivity simulations mentioned before.

   By employing Bayes' formula the extreme occurrence probabilities can be expressed with conditional probabilities as

$$p_{0/1} = \mathrm{Pr}(T_{0/1} \lessgtr T_{\mathrm{ref}}|F_{0/1} \in \mathcal{C}_{\mathrm{ref}}) \cdot \frac{\mathrm{Pr}(F_{0/1} \in \mathcal{C}_{\mathrm{ref}})}{\mathrm{Pr}(F_{0/1} \in \mathcal{C}_{\mathrm{ref}}|T_{0/1} \lessgtr T_{\mathrm{ref}})} \tag{2}$$

Here, $\mathcal{C}_{\mathrm{ref}}$ describes the set of all slp anomaly fields or atmospheric flows $F_{0/1}$ in the respective world that are allocated to a certain reference regime centroid $C_{\mathrm{ref}}$. When applying this decomposition we assume that the storyline of an extreme at a specific grid point can be explained by the presence of a specific reference regime $C_{\mathrm{ref}}$.

   The probability or risk ratio $\rho$ compares the extreme occurrence probabilities in the counterfactual ($p_0$) and in the factual world ($p_1$). When using Eq. 2 this ratio can be multiplicatively decomposed into

$$\rho = \frac{p_0}{p_1} = \rho_{FR} \cdot \rho_{CR} \tag{3}$$





that is a term $\rho_{\mathrm{CR}}$ ("Changed-Regime") relating changes in extremes to changes in regime occurrences, and another term $\rho_{\mathrm{FR}}$ ("Fixed-Regime") that considers such changes in extremes by fixing a certain circulation regime.

The Fixed-Regime contribution term is given by

$$\rho_{\mathrm{FR}} = \frac{\Pr(T_0 \lesseqgtr T_{\mathrm{ref}} | F_0 \in \mathcal{C}_{\mathrm{ref}})}{\Pr(T_1 \lesseqgtr T_{\mathrm{ref}} | F_1 \in \mathcal{C}_{\mathrm{ref}})} \tag{4}$$

This contribution term describes the extreme occurrence probability ratio between both worlds given a regime allocation $F_{0/1} \in \mathcal{C}_{\mathrm{ref}}$ to a certain reference regime set $\mathcal{C}_{\mathrm{ref}}$. This terms has previously been named thermodynamical contribution (Yiou et al., 2017), as the atmospheric circulation is fixed in terms of circulation regimes. Nevertheless, caution is needed when using such names as this term to a certain extent assumes that the regime pattern structures do not change between simulation scenarios. For weak forcings this has however been shown to be a valid assumption (Palmer, 1999) (see also Figure A2
for comparison of different pattern structures computed for different combinations of simulations). In addition to this, the individual flows allocated to a respective set $\mathcal{C}_{\mathrm{ref}}$ may also differ between different simulations.

The second contribution related to regime changes is defined as

$$\rho_{\mathrm{CR}} = \rho_{\mathrm{reci}} \cdot \rho_{\mathrm{circ}} = \frac{\Pr(F_1 \in \mathcal{C}_{\mathrm{ref}} | T_1 \lesseqgtr T_{\mathrm{ref}})}{\Pr(F_0 \in \mathcal{C}_{\mathrm{ref}} | T_0 \lesseqgtr T_{\mathrm{ref}})} \cdot \frac{\Pr(F_0 \in \mathcal{C}_{\mathrm{ref}})}{\Pr(F_1 \in \mathcal{C}_{\mathrm{ref}})}. \tag{5}$$

The latter term $\rho_{\mathrm{circ}}$ is related to changes in the occurrence probability of the reference regime $C_{\mathrm{ref}}$ between both simulations.
Therefore, this term has previously also been termed dynamical contribution (Yiou et al., 2017), as $\rho_{\mathrm{circ}}$ can be directly associated to dynamical changes within the framework of circulation regimes. The term $\rho_{\mathrm{reci}}$ evaluates changes in the probability of a circulation such as $C_{\mathrm{ref}}$ when given an extreme. $\rho_{\mathrm{reci}}$ allows for connecting the more meaningful and interpretable terms $\rho, \rho_{FR}, \rho_{\mathrm{circ}}$ and it has also been suggested by Yiou et al. (2017) that it helps to reconcile the risk-based approach (estimation of $\rho$ only) with the storyline approach.

**3.3  Uncertainty estimates**

Uncertainty and significance estimates are reported with confidence intervals based on the 0.05 and 0.95 quantile of bootstrapped distributions of the relevant statistic. If the computed confidence intervals do not include unity (for ratios) or a zero value (for differences) the signal is termed significant. Daily temperature time series, as well as daily nominal time series of cluster allocations typically exhibit significant temporal dependencies over several days. In order to preserve the temporal
structure of the original daily data during the resampling procedure a moving block bootstrap is used here (Kunsch, 1989).

The original time series $x_n$ of length $n$ is therefore divided into overlapping blocks of size $k$, where the first Block contains $x_1, ..., x_k$ and the second block $x_2, ..., x_{k+1}$ etc. . Afterwards, a bootstrap sample is created by concatenating randomly picked blocks to a new time series of original length $n$ and the statistic of interest (cluster frequency, $\rho$ etc.) is computed for this generated bootstrap sample time series. When employing this procedure for statistics where multiple variables are involved
(e.g. $\rho_{\mathrm{reci}}$ and $\rho_{\mathrm{FR}}$), the time series of temperatures and regime allocations are blocked and resampled pairwise. This procedure is repeated 1000 times yielding a bootstrapped probability distribution of the respective statistic of interest. The block length $k$ is set to 5 days corresponding to a typical persistence time of the circulation regimes.





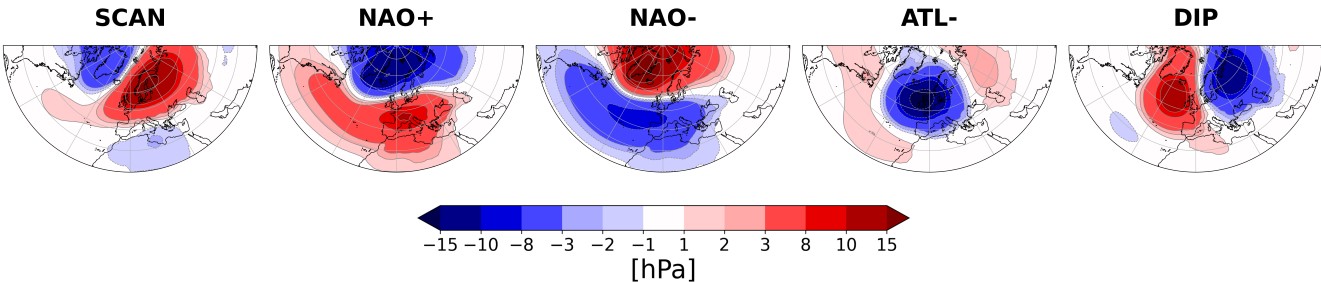

**Figure 1.** Five circulation regimes over the Euro-Atlantic domain computed from daily ECHAM6 PAMIP slp anomaly data for the extended winter season (December, January, February, March). For the displayed regimes, data from the reference simulation and the futArcSI simulation have been merged before applying the clustering algorithm.

## 4    Results and Discussion

In the upcoming section we present results of the analysis steps already outlined in the introduction. Initially, the impact of
Arctic sea ice changes on the large-scale atmospheric winter circulation is assessed within the context of atmospheric circulation regimes. Therefore, we compare results obtained from model and ERA5 reanalysis data and discuss how future Arctic sea ice changes impact the occurrence probability of certain circulation regimes. Subsequently we demonstrate how winter temperature extremes over Europe can be associated with certain circulation regimes. Based on the previous analysis steps and after discussing to what extent overall changes in winter temperature extremes can be observed in the sea ice sensitivity
simulations, we assess how these changes over the European domain can be related to circulation and non-circulation related contributions.

### 4.1    Regime frequency changes induced by future sea ice retreat

To start with we discuss how the occurrence frequency of the computed atmospheric circulation regimes is affected by future Arctic sea ice changes. Fig. 1 shows the five circulation regimes computed for the extended winter season over the Euro-
Atlantic domain (-90°W–90°W, 20°N–88°N). Daily slp anomaly data merged together from the pdSI/pdSST and the futArcSI simulation data have been used here. The computed regimes closely resemble regimes found in previous studies (e.g. Crasemann et al., 2017) and include a frequently detected Scandinavian Blocking regime (Dorrington and Strommen, 2020; Falkena et al., 2020; Yiou et al., 2017), termed SCAN, with an anticyclonic blocking structure over Scandinavia and parts of the Ural mountains. As shown in Fig. A3a, up to 40% of SCAN regime days are indeed accompanied by blocking activity over northern
and northeastern Europe. Studies by for instance Jung et al. (2017) and Sato et al. (2014) showed that such an anticyclonic anomaly over northeastern Europe might be part of a wave train structure that originates from the east coast of North America and is forced by warming anomalies over this remote region. Another regime is characterized by a cyclonic structure over the Atlantic and parts of western Europe (ATL-) and has previously been named negative Atlantic ridge (Falkena et al., 2020) or Scandinavian trough (Dorrington and Strommen, 2020). A dipole pattern (DIP) is found with positive pressure anomalies





over the North Atlantic and negative pressure anomalies over northeastern Europe that has also been frequently termed Atlantic
Ridge (Dorrington and Strommen, 2020; Falkena et al., 2020; Yiou et al., 2017). Finally, two of the computed regimes resemble
the positive (NAO+) and negative (NAO-) phase of the North Atlantic Oscillation, respectively.

The structure of the individual regimes is relatively unaffected by the exact definition of winter season (e.g. by excluding
March) and by modifications of the spatial domain (using e.g. -80°W-80°W, 30°N-88°N). Compared to circulation regimes

computed from ERA5 data (see Fig. A1), it appears that ECHAM6 is able to realistically simulate the spatial structure of these
five regimes. Indeed, Fig. A2 indicates high spatial correlations (generally greater than 0.9), and similar (but e.g. for NAO+
slightly higher) pattern amplitudes when comparing regimes computed from different combinations of model simulations with
ERA5 regimes. The fact that ECHAM6 is able to realistically simulate these large scale circulation features allows for a
reasonable comparison with ERA5 in terms of regime occurrence frequency changes.

In order to assess the impact of future Arctic sea ice changes on the occurrence probability of certain regimes, Fig. 2 shows
monthly-splitted histograms comparing the relative occurrence frequencies of the computed regimes between the reference
simulation and the futArcSI (Figs. 2a-e), as well as with the futBKSI (Figs. 2f-j) sea ice sensitivity experiment. Comparing
these modeled frequency changes with tendencies in ERA5 allows for supporting the robustness of the detected model signals
(see triangles in Fig. 2 for comparison between months with above and below average Arctic sea ice area over the period

1979–2018). Overall, it can be observed that the regime changes in Fig. 2 associated with the futArcSI and futBKSI sensitivity
simulations share many similar features. Consistent with previous studies this again emphasizes the potential key role of sea
ice loss in the Barent/Karasea region when trying to identify and understand linkages between the Arctic and mid-latitudes.
An overall midwinter increase of SCAN occurrence by several percent is detected in both sea ice sensitivity simulation as well
as in the reanalysis data for low sea ice conditions (see Figs. 2a and f) . For futBKSI this frequency change is significantly

pronounced in January and February, whereas for futArcSI the signal is only detectable in January. Such an overall midwinter
SCAN response is consistent with previous studies, such as by Luo et al. (2016) who related a strengthening of the Scandinavian
or Ural Blocking in winter season to instantaneous sea ice loss in the Barent/Karasea region. Petoukhov and Semenov (2010)
analyzed model simulations and showed that for moderate winter sea ice reductions over the Barent/Karasea an anticyclonic
anomaly centered over the same region can be observed in February; however, they emphasized that such an anticyclonic

circulation response depends on the actual prescribed magnitude of sea ice loss in the Barent/Karasea in a highly nonlinear
way. Within the framework of circulation regimes Crasemann et al. (2017) detected an increased December SCAN occurrence
frequency—however only in response to recent Arctic sea ice loss. It should be mentioned that a variety of recent modeling
studies (Kim et al., 2022; Peings, 2019) did not find any intensifications of Ural blockings in response to sea ice loss over the
Barent/Karasea region.

In addition to the previously discussed changes in SCAN occurrences, especially the futBKSI sensitivity simulation reveals
a decreased occurrence frequency of the NAO+ and NAO- pattern in February (Figs. 2g and h). This might be interpreted as
a weakened dominance of NAO variability under future conditions. However, only the diminished occurrence frequency of
the NAO+ pattern can be observed in the reanalysis as well. Such a reduction of positive NAO events is consistent with the
commonly reported negative NAO response (Jaiser et al., 2012; Screen, 2017b; Nakamura et al., 2015; Deser et al., 2010) to





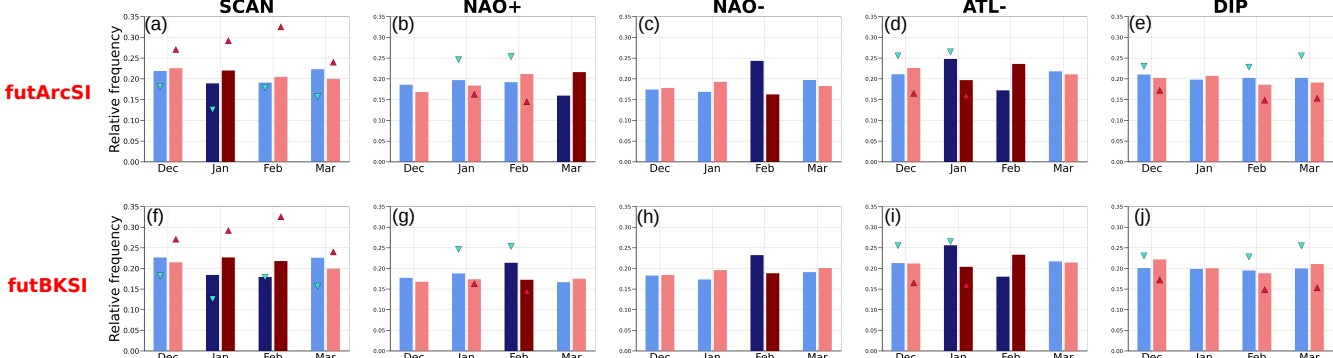

**Figure 2.** Relative regime occurrence frequencies for different winter months compared between the pdSST/pdSI reference simulation (blueish bars) and the futArcSI (upper row, redish bars), as well as the futBKSI sensitivity simulation (lower row, redish bars). Transparent redish and blueish bar indicate non-significant frequency differences between reference and sensitivity simulations, whereas the paired dark blueish/redish bars indicate significant differences in occurrence frequencies. Note that by definition the sum over all clusters for a specific month in a given simulation is one. The triangles indicate the respective ERA5 regime occurrence frequencies for low (upright redish triangles, lower 50% of multiyear mean) and high (inverted bright-blueish triangles, upper 50% of multiyear mean) Arctic sea ice conditions derived from linearly detrended monthly Arctic sea ice area data over the period 1979–2018. Only ERA5 occurrence frequencies for months where significant differences between low and high Ice conditions were found are shown here. Significant differences are derived from a moving block bootstrap.

sea ice loss. Another significant signal found in both, reanalysis and model simulations, is a more frequent occurrence of the ATL- pattern in January under higher sea ice conditions in the reference simulation (Figs. 2d and i).

## 4.2   Links between certain circulation regimes and European temperature extremes

After examining how the occurrence probability of certain circulation regimes can be affected by sea ice changes we now discuss which of the computed circulation regimes can be associated with temperature extremes over Europe. For this reason,

Fig. 3 compares the occurrence probability of temperature extremes given a specific circulation regime to the unconditioned probability of an overall extreme occurrence. Although this is only shown here for the pdSST/pdSI reference simulation, results when using data from the sensitivity model experiments and even for ERA5 are qualitatively extremely similar. Figure 3c indicates that the presence of a NAO- regime is associated with an up to more than doubled probability than usual of cold extreme days over large parts of mid- to northern Europe. This observed link between NAO- events and winter cold spells or

negative temperature anomalies over northern Europe is well-established and frequently observed in studies (Cattiaux et al., 2010; Andrade et al., 2012; Rust et al., 2015; Screen, 2017b). Figure A5c shows how NAO- event are related to easterly zonal wind anomalies which consequently lead to favored cold air advection of continental air masses towards northern Europe. These easterly anomalies can generally also be related to a suppressed advection of warmer maritime air masses, favoring colder conditions over Europe. As shown in Fig. A3b, up to 40% of NAO- regime days are associated with atmospheric



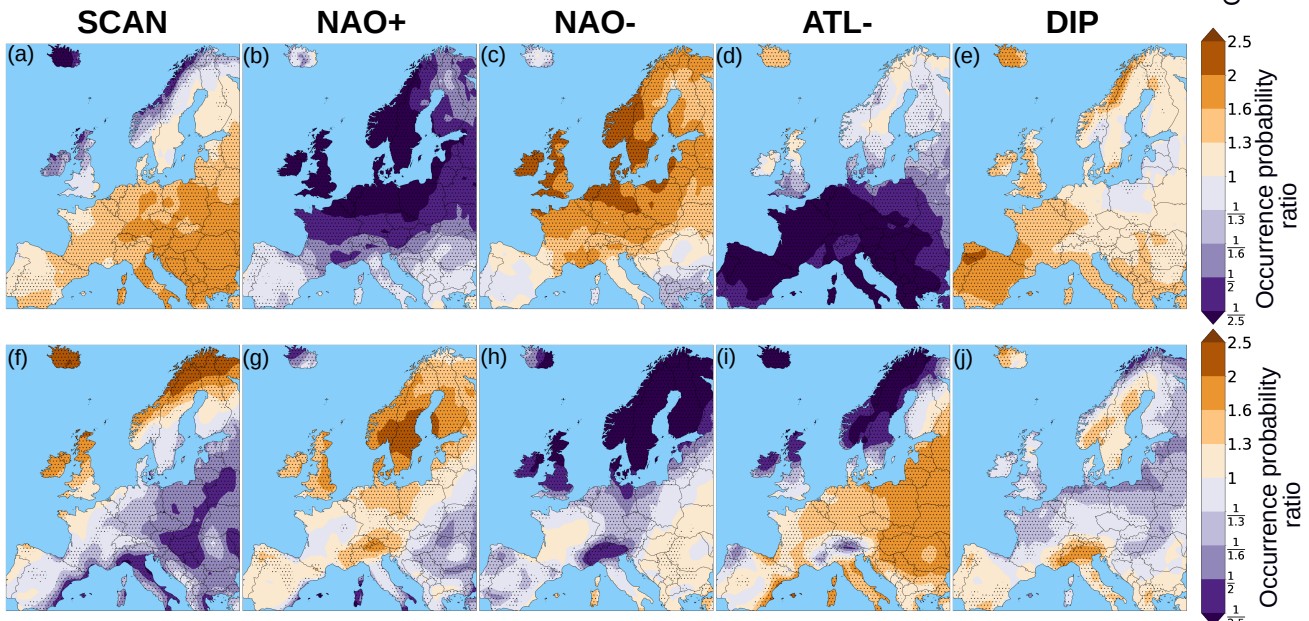

**Figure 3.** Temperature extreme occurrence probability ratios (DJFM) for different circulation regimes plotted over the European domain using the ECHAM6 PAMIP pdSST/pdSI simulation. Upper row a-e: cold days, bottom row f-j: warm days. The plotted ratio compares the occurrence probability of an extreme day given a certain circulation regime to the unconditioned probability of an extreme occurrence. Thus, values greater than one at a specific grid point indicate preferred extreme occurrence during the presence of a certain regime compared to the overall extreme occurrence. Dotted areas indicate ratios that are significantly different from unity based on a moving bootstrap.

blocking activity over Greenland and the North Atlantic. Blocking conditions over these region have previously been related to European winter cold spells as well (Sillmann et al., 2011).

In addition to the NAO- regime preferred occurrences of cold extremes over central and eastern Europe can be observed during SCAN days in Fig. 3a. Links between anticyclonic systems over Scandinavian/Ural regions and cold days over large parts of Europe have been reported previously (Petoukhov and Semenov, 2010; Andrade et al., 2012), since Scandinavian high

pressure system are typically associated with cold air advection towards central Europe from northeastern European regions (see Fig. A5a). Indeed, Lagrangian backward trajectory analyses (Bieli et al., 2015) showed that cold events over mid- and eastern Europe are induced by horizontal advection of air masses from Russia and far northeastern regions. These advective processes are furthermore characterized by an adiabatic and steady descend of the air masses. Additionally, Fig. 3e indicates preferred cold extreme occurrences over most parts of western Europe during the presence of the Dipole regime. This link

is related to southward advection (see Fig. A5e) of Arctic air masses especially from regions east of Greenland (Bieli et al., 2015).

Warm days in winter over large parts of central, eastern and southern Europe occur preferably during the presence of the ATL- regime (see Figure 3i). As shown in Fig. A4a, around and westwards of the British Isles the ATL- regime is associated



with enhanced baroclinic activity and consequently an intensification of the North Atlantic storm track. Therefore, more storm
systems than usual may form and advect warm and moist Atlantic air masses towards mid- and southern Europe. Comple-
mentary, warm days over northern Europe are linked to the presence of the NAO+ regime (see Fig. 3g). Such warm extremes
over northern Europe are linked to strengthened westerly transport of moist Atlantic air masses during positive NAO events
resulting in enhanced latent energy transport towards Scandinavia (Vihma et al., 2020). As shown in Fig. A4b this can also be
related to a poleward shift of the North Atlantic stormtrack towards northern Europe and the Arctic.

**4.3    Sea ice induced changes in winter temperature extremes**

Now focus shifts to the question what changes in winter temperature extremes over continental parts of the Northern Hemi-
sphere can be expected in response to future sea ice loss. Therefore, Figs. 4 and 5 depict the overall occurrence ratio $\rho$ of cold
and warm extremes, comparing the extreme occurrence probability in the futArcSI and futBKSI experiments with the reference
simulation. Figure 4 indicates a general tendency towards less frequent cold extreme occurrences in the future sea ice scenario
simulations over the mid- to high northern latitudes. From a thermodynamical perspective this observation is consistent with
the fact that more open water areas and the associated elevated surface temperatures in the sensitivity runs provide an addi-
tional energy source to the atmosphere. However, the spatial pattern and the signals' magnitude strongly depend on the specific
month and whether sea ice is reduced over the entire Arctic (see Fig. 4e-h) or just over the Barent/Karasea (see Fig. 4a-d).
Although spatial tendencies show to some extent relatively similar patterns in both sensitivity simulations, futArcSI exhibits
much more pronounced reductions in cold extremes by a factor of more than 2.5 over high northern latitudes. Contrary, some
parts over mid- and northern Eurasia show more frequent cold extreme occurrences in futBKSI from January to March. This
observation is consistent with the frequently reported Eurasian cooling response to sea ice loss in the Barent/Karasea (Cohen
et al., 2018) that has been associated with a strengthening of the Siberian high. Over Europe significant reductions of cold
extreme occurrences can be observed in futBKSI in February (Fig. 4c), as well as in the futArcSI simulation in February and
March (Figs. 4g and h). Interestingly, January tends to exhibit slightly more cold extremes over central and eastern Europe in
both sensitivity simulations (Figs. 4b and f).

As illustrated in Fig. 5 significant changes in the occurrence of warm extremes are generally less pronounced compared
to cold extremes. Over Europe an overall tendency towards more frequent occurrences of warm extremes can be detected
especially under diminished Arctic sea ice conditions in the futArcSI simulation (Figs. 5e-h). In many regions and month
reductions in cold extreme occurrences are accompanied by increased probabilities of warm extremes. This might be associated
with an overall thermodynamical shift of the underlying temperature distribution due to reduced sea ice concentrations and
warmer surface temperatures in the sensitivity experiments. For futArcSI this is e.g. the case over northern Siberia in December
(Figs. 4e and 5e) or over Europe in March (Figs. 4h and 5h). However, several regions such as central Europe in February show
for instance in futBKSI reductions in cold extreme occurrences but no significant complementary changes in warm extremes
(see Figs. 4c and 5c). Such asymmetric responses in the tails of the temperature distributions can not be explained by simple
thermodynamical arguments and are certainly a result of other contributing factors such as changes in the dynamical situation
leading to a certain extreme. In rare cases such as over central and eastern Europe in January, the futArcSI experiment even





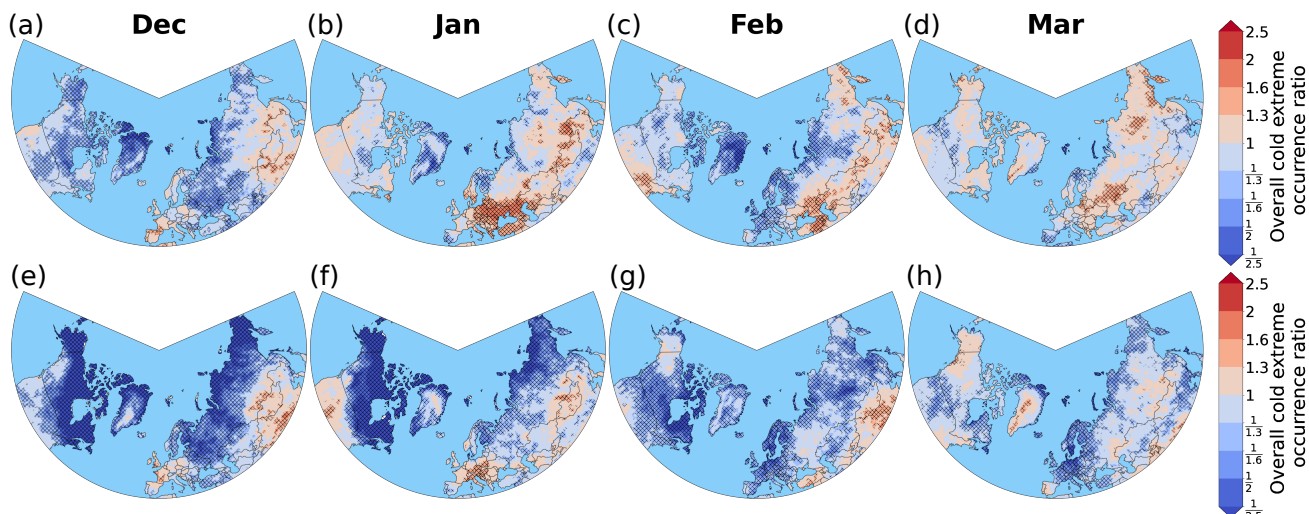

**Figure 4.** Cold extreme occurrence ratio for December, January, February and March. The occurrence probability of northern hemispheric continental cold extremes are compared between the sensitivity experiments (more frequent occurrences red) vs. the pdSST/pdSI reference simulation (more frequent occurrences blue). Upper row (a)–(d): futBKSI sensitivity run. Bottom row (e)–(h): futArcSI sensitivity run. Hatching indicates regions where the ratio differs significantly from unity based on a moving block bootstrap.

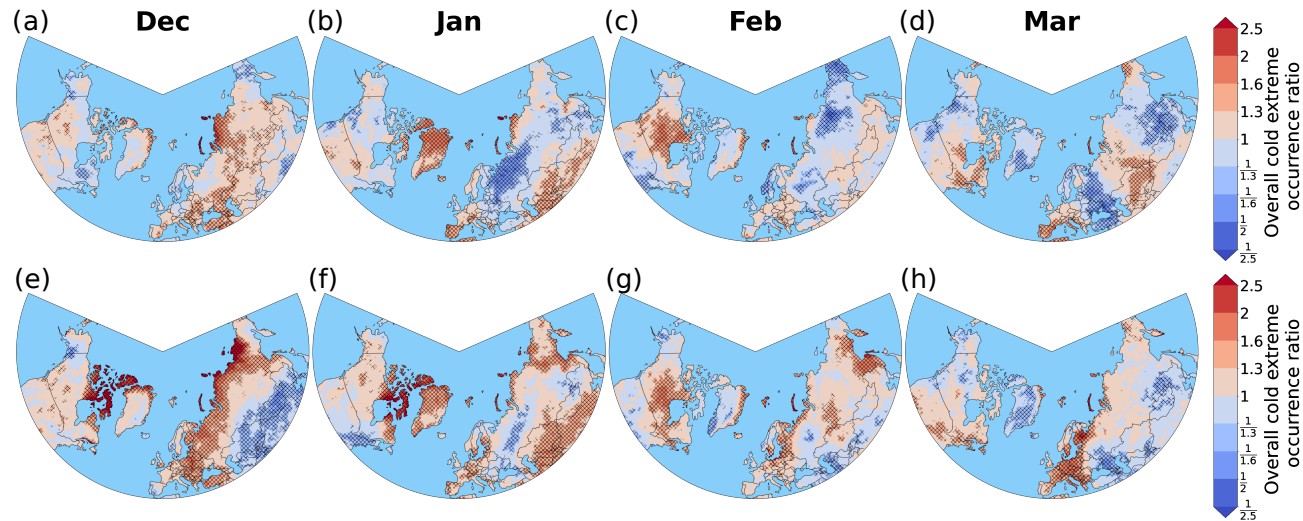

**Figure 5.** Same as in Fig. 4 but for warm extremes.

shows an increased occurrence probability of both, cold and warm extremes (see Figs. 4f and 5f). This might be also related to an overall increase of temperature variability.



**Figure 6.** Conditional extreme event attribution framework for European cold extremes assuming a SCAN-storyline. Compared are the pdSST/pdSI reference simulation (blue indicates favored occurrence) and the futBKSI sensitivity simulation (red indicates favored occurrence). Upper row: January with $\rho_{\mathrm{circ}} = 1.26$. Bottom row: February with $\rho_{\mathrm{circ}} = 1.23$. $\rho_{\mathrm{circ}}$ greater than unity means that the SCAN regime occurs more frequent in the futBKSI simulation for both months (see also Fig. 2f). The first column shows the overall cold extreme occurrence ratio $\rho = \rho_{\mathrm{FR}} \cdot \rho_{\mathrm{CR}}$ between both simulations, the second column shows the Fixed-Regime contribution $\rho_{\mathrm{FR}}$, the third one shows the Changed-Regime contribution $\rho_{\mathrm{CR}}$. Stippling indicates regions where the ratios significantly differ from unity based on a moving block bootstrap. For regions colored in white the respective terms could not be calculated as no matches between extreme and circulation regime occurred.

## 4.4 Decomposition of extreme frequency changes

Now focus finally shifts back to temperature extremes over Europe. We try to understand to what extent changes in extreme occurrences over Europe can be explained by sea ice induced circulation and non-circulation related changes. Therefore, we now employ the conditional extreme event attribution framework described in Sect. 3.2 and compute the Fixed-Regime





contribution term $\rho_{FR}$, as well as the Changed-Regime contribution term $\rho_{CR}$. On the one hand, we showed in Sect. 4.1 how

the occurrence probabilities of the SCAN, NAO+ and ATL- pattern in January and/or February are significantly affected by future sea ice reductions. On the other hand, in Sect. 4.2 we discussed how these regimes can be statistically and dynamically related to preferred occurrences of European temperature extremes. Based on these results, the following decompositions of overall responses in extreme occurrences are considered here for the futBKSI simulation: European cold extremes along a SCAN storyline in January and February, warm extremes along a ATL- storyline in January, as well warm extremes along a

NAO+ storyline in February. Only months for which robust and significant changes in regime occurrence frequencies have been detected (see Sect. 4.1) are considered here, since the physical interpretation of the Changed-Regime term $\rho_{CR}$ strongly relies on significant changes in $\rho_{circ}$. Furthermore, decomposition plots are shown over the entire European domain; however, for interpretation it should be kept in mind that specific regimes are only related to extremes over certain parts of Europe. Results for the futArcSI simulation are shown in the Appendix and are also discussed below.

Figure 6 shows the decomposition of the overall cold extreme occurrence ratio $\rho$ between the pdSST/pdSI reference simulation and the futBKSI sensitivity simulation for January (Figs. 6a-c) and February (Figs. 6d-f). The SCAN regime was chosen as the reference pattern $C_{ref}$ since it could be associated with cold extremes over central, western and eastern Europe (Fig. 3a) and revealed significant frequency changes in the midwinter months as well (Fig. 2f). In January it shows that eastern and parts over central Europe are associated with significantly more frequent cold extremes in the futBKSI simulation (Fig. 6a).

The decomposition reveals that these signals can especially over central Europe be associated with a significant contribution of the Changed-Regime $\rho_{CR}$ term (Fig. 6c). This contribution is related to a 26% increase of SCAN regime occurrences in the futBKSI simulation in January (see also Fig. 2f). Such a dynamical contribution is however absent in more eastern parts of Europe, where the Fixed-Regime term $\rho_{FR}$ significantly contributes (Fig. 6b). In February, strong frequency decreases of cold extremes over large parts of western, central and northern Europe can be observed in the futBKSI simulation (Fig. 6d).

In contrast to January, the predominant part of these overall changes is explained by the Fixed-Regime term $\rho_{FR}$ (Fig. 6e). This might be interpreted as an overall thermodynamical warming effect since more ice-free areas in the model simulations are typically associated with warmer surface temperatures and with overall stronger ocean–to–atmosphere heat fluxes. Such additional heat and energy sources provided to the atmosphere are finally distributed via the climatological mean circulation. As air masses from northeastern Europe and the Barent/Karasea frequently serve as source regions for advective processes

leading to cold spells over central Europe (Bieli et al., 2015), an average warming of these reservoir regions may suppress the occurrence of cold extremes over Europe in the futBKSI simulation. As it can be seen for the Changed-Regime term $\rho_{CR}$ in Fig. 6f, February frequency changes in SCAN occurrences basically tend to favor cold extremes over most parts of Europe. However, compared to the Fixed-Regime term $\rho_{FR}$ (Fig. 6e) these signals are relatively small and non-significant over most areas.

The same analysis for January is illustrated in Fig. A6 but considers the futArcSI simulation instead of the futBKSI simulation. The overall cold extreme response (Fig. A6a) shows a significantly increased (decreased) probability of cold extreme occurrences over some parts of central (northeastern) Europe. The increased cold extreme probability over central Europe in Fig. A6a shows how two non-significant contributions (Figs. A6b and c) may add up to a significant overall response, whereas





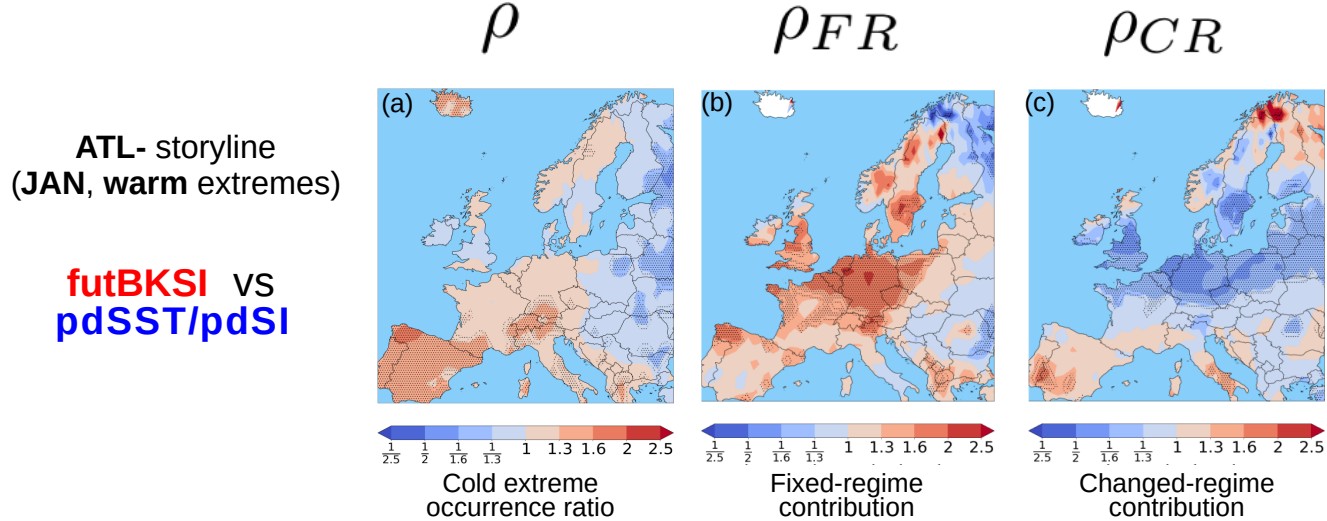

**Figure 7.** Same as in Fig. 6 (BKSI vs. pdSST/pdSI simulation) but for January warm extremes and along a ATL- regime storyline. Occurrence ratio of ATL- regime occurrence in January is given as $\rho_{\mathrm{circ}} = 0.8$. Thus, the ATL- occurs less frequent in the futBKSI simulation (see also Fig. 2i).

the decreased cold extreme probability over northeastern Europe is mostly explained by the Fixed-Regime term $\rho_{\mathrm{FR}}$ (Fig. A6b).

Figure 7 shows the decomposition for European warm extremes in January by considering the ATL- regime as the reference pattern $C_{\mathrm{ref}}$. Here, the non-presence of significant signals in the overall warm extreme occurrence ratio over most parts of Europe (Fig. 7a) is especially over mid- and parts of eastern Europe a result of opposing $\rho_{\mathrm{FR}}$ (Fig. 7b) and $\rho_{\mathrm{CR}}$ (Fig. 7c) contributions. On the one hand, the reduced ATl- occurrence in the futBKSI simulation can be associated with less frequent advections of warm air masses by Atlantic storm systems. On the other hand, an overall thermodynamical warming effect as mentioned before due to more open water areas tends to favor the occurrence of warm extremes. A similar line of reasoning for January warm extremes along a ATL- storyline can be used in Fig. A7 where the futArcSI simulation is considered and both contributions also appear to counteract each other. Here, an overall tendency towards more warm extremes can be observed over several parts of Europe compared to the futBKSI simulation. This stems from a stronger dominance of the Fixed-Regime term $\rho_{\mathrm{FR}}$ (Fig. A7b) probably due to a more pronounced thermodynamical forcing for Arctic-wide sea ice loss compared to sea ice loss over the Barent/Karasea only.

Figure 8 shows the decomposition for European warm extremes in February. The NAO+ regime is considered here as the reference pattern $C_{\mathrm{ref}}$, since, on the one hand it can be associated with warm extremes especially over more northern parts of Europe. On the other hand it showed significantly less frequent occurrences in the futBKSI simulation in February. The overall warm extreme occurrence ratio $\rho$ only shows some significantly less frequent extreme occurrences in the futBKSI simulation





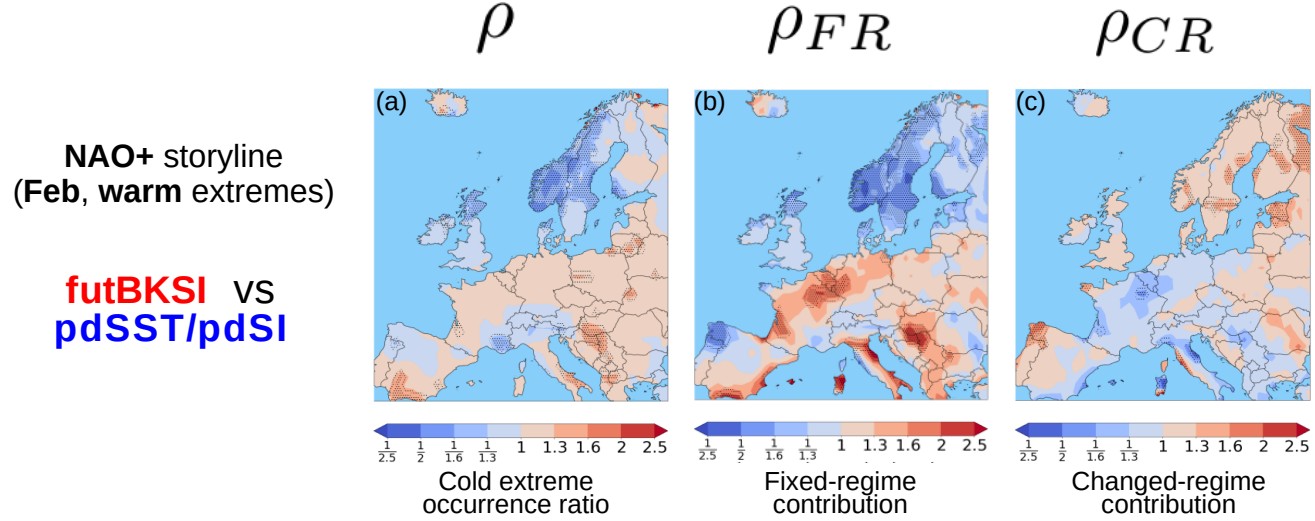

**Figure 8.** Same as in Fig. 6 (BKSI vs. pdSST/pdSI simulation) but for February warm extremes and assuming a NAO+ regime storyline. Occurrence ratio of NAO+ regime occurrence in February is given as $\rho_{\mathrm{circ}} = 0.8$. Thus, the NAO+ regime occurs less frequent in the futBKSI simulation (see also Fig. 2g).

over parts of Scandinavia (Fig. 8a). These signals are mostly explained by the Fixed-Regime contribution $\rho_{\mathrm{FR}}$ in Fig. 8b. The term $\rho_{\mathrm{CR}}$ shows basically no significant contribution (Fig. 8c).

Finally, the previous results are contrasted to results for global SST changes in order to assess the relative importance of Arctic sea ice loss compared to a future increase of global SSTs. Therefore, Fig. 9 shows the overall response and the two contributions $\rho_{\mathrm{FR}}$ and $\rho_{\mathrm{CR}}$ for midwinter cold extremes comparing the reference and the futSST simulation. The NAO- pattern was set as the reference pattern here, but results for other storylines reveal the same qualitative picture for. First, it shows that cold extremes occur massively and significantly less frequent in the futSST simulation over all parts of Europe (Fig. 9a). Secondly, these overall changes are almost completely explained by the fixed-circulation term $\rho_{\mathrm{FR}}$ (Fig. 9b). Although the NAO- regime only shows non-significant changes between both simulation in this case ($\rho_{\mathrm{circ}} = 0.96$), even significant and more distinct changes in regime occurrences could not contribute in the same way as the Fixed-Regime contribution. This illustrates how the overall thermodynamical warming effect induced by warmer global SSTs clearly dominates any circulation induced changes in extreme occurrences. A similar pictures is found for warm extremes. Therefore, we can conclude that although sea ice is able to affect extreme occurrences over Europe via dynamical and thermodynamical contributions, compared to future SST and for sure also to future global warming the effect is rather small.



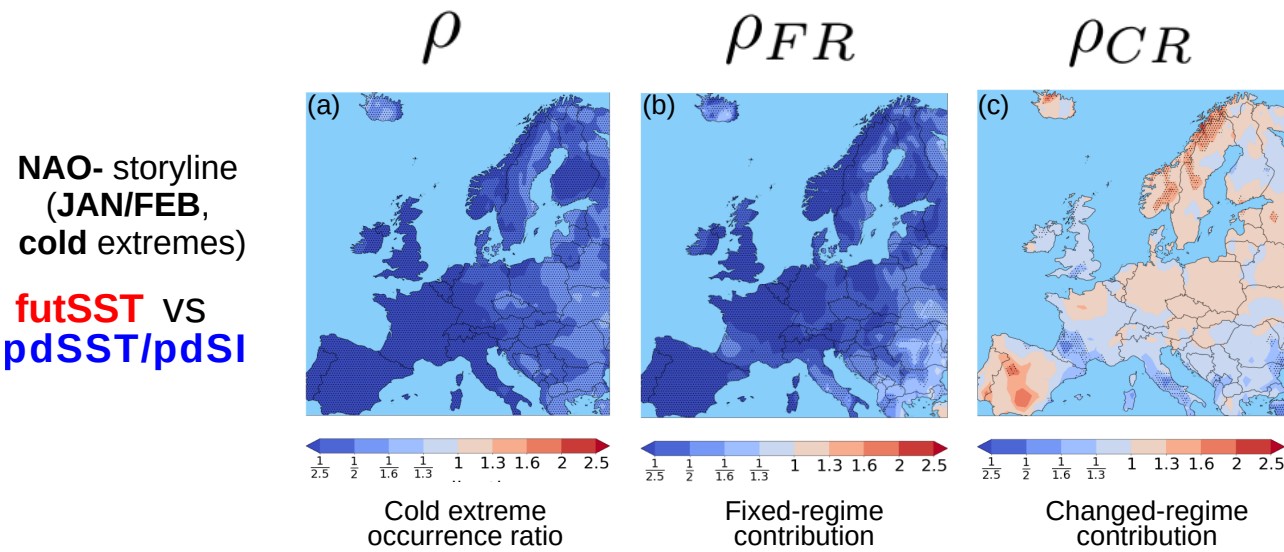

**Figure 9.** Similar to Fig. 6, but comparing the pdSST/pdSI reference simulation (blue indicates favored occurrence) and the futSST sensitivity simulation (red indicates favored occurrence). Analyzed are cold extremes in January/February and a NAO- storyline is assumed here.

## 5 Summary

The general aim of this paper was to discuss how future Arctic sea ice retreat is able to impact the occurrence frequency of temperature extremes over Europe. Therefore, for the most part we investigated data from ECHAM6 sea ice sensitivity model experiments that are part of the PAMIP data pool. We considered simulations forced under future sea ice reduction over the entire Arctic, as well as only over the Barent/Karasea and compared them to a sensitivity simulation forced under present day conditions.

We initially studied how such future sea ice reductions can affect the large-scale circulation over the Euro-Atlantic region in terms of occurrence frequency changes of atmospheric circulation regimes. Therefore, five Euro-Atlantic circulation regimes have been computed with $k$-means clustering. Afterwards, we discussed which circulation regimes can be associated with cold or warm extremes over Europe and how the prescribed sea ice loss in the sensitivity simulations can impact the occurrence frequency of such extremes over the Northern Hemisphere. Based on the previous analysis steps, we employed a framework of conditional extreme event attribution and decomposed the overall extreme changes over Europe along suitable regime storylines. This was done in order to understand the different thermodynamical and dynamical factors in a better way that may contribute to the modeled overall responses in extremes. The decomposition of changes in extreme event frequencies finally yielded respective contributions related to changes in the circulation (changes in regime occurrences), and contributions assuming fixed atmospheric dynamics in terms of circulation regimes.

The findings of the different analysis steps and research questions mentioned in the beginning can be summarized as follows:



– *Within the methodological framework of atmospheric circulation regimes, what changes in the wintertime atmospheric large-scale circulation over the Euro-Atlantic sector can be expected under future Arctic sea ice retreat?* As already motivated by Crasemann et al. (2017), we also detected a variety of significant changes in the occurrence frequency of winter circulation regimes when contrasting idealized atmosphere-only model simulation forced under present day and future Arctic sea ice conditions. However, the sign and significance of the signals highly depend on the respective month. Furthermore, sea ice reduction locally prescribed only over the Barent/Karasea already explained most of the frequency changes found for Arctic-wide sea ice loss. When comparing modeled signals with tendencies of reduced sea ice conditions found in ERA5, we found a general increase of Scandinavian blocking (SCAN) occurrences in midwinter, as well as a decrease of a cyclonic pattern over the eastern Atlantic (ATL-) in January. For sea ice loss only prescribed over the Barent/Karasea we also detected a consistent reduction of NAO+ occurrences in February.

– *Which regimes can be associated with preferred occurrences of winter temperature extremes over Europe?* We showed and discussed that cold (warm) extremes over southern, central and eastern Europe occur significantly more frequent during SCAN (ATL-) days, whereas especially cold extremes over central- to northern Europe are on average significantly more frequently associated with negative (positive) NAO regime events.

– *What overall frequency changes of extreme occurrences over the continental Northern Hemisphere can be detected in response to future sea ice changes?* We found that prescribed sea ice reductions in the model simulation resulted in an overall tendency towards less cold extreme days, especially over high northern continental regions. A general tendency towards more warm extremes was less clear. However, the signal structures, their signs as well as their significances highly depend on the specific region and month. Finally we noticed that reductions in cold extreme occurrences are not necessarily accompanied by less frequent occurrences of warm extremes, and vice versa.

– *Based on the sea ice induced changes in circulation regimes, to what extent can frequency changes of European extremes be related to circulation and non-circulation related contributions?* The decomposition of overall responses of midwinter extreme occurrences revealed a rather complex picture. In several cases we could associate significant contributions related to occurrence frequency changes of certain regimes to preferred or unfavored occurrences of extremes. This was especially the case for increased January cold extremes related to increased Scandinavian Blocking occurrences, or decreased January warm extremes related to a reduced frequency of the ATL- pattern. Furthermore, we observed in several cases that the contribution related to fixed circulation regimes yielded from a thermodynamical point of view intuitively expected decreased (increased) occurrence frequencies of cold (warm) extremes under future sea ice conditions. Finally, we noticed different scenarios for the resulting overall extreme occurrence frequency response. First, one contribution may dominate and results in a significant overall response. This was for instance the case for February cold extremes under a SCAN storyline where the overall reduced extreme occurrence frequency is explained by the Fixed-Regime contribution. Secondly, changes in regime occurrences may counteract the general thermodynamical warming or cooling trend resulting in no detectable overall change in extreme occurrences. This was especially observed for January warm extremes under a ATL- storyline.



When analyzing changes in midwinter cold extremes induced by future raised global SSTs we observed a strong and significant decrease of cold extremes occurrences over entire Europe, especially when contrasted to results obtained for future sea ice reductions. Furthermore, this decrease was nearly completely explained by the contribution related to fixed atmospheric dynamics. This suggests a dominance of thermodynamical warming arguments over changes in atmospheric dynamics when 495 trying to understand future changes in European temperature extremes. Overall these observations indicate that although future Arctic sea ice loss is for sure able to affect temperature extremes over Europe and the related atmospheric dynamics, the total effect size compared to globally raised temperatures that are expected in the future is relatively small.

## 6   Concluding remarks

Some limitations, also in terms of interpretation may arise from the specific model setup and methodology that is used here. The 500 question to what extent the detected winter changes in extremes or circulation regimes are a result of time-delayed stratospheric pathways triggered by sea ice loss in autumn cannot be answered with the presented methodology and experimental design. From the experimental side this would require more tailored model experiments as for instance done by Blackport and Screen (2019). They compared the delayed effect of autumn and year-round sea ice loss on the winter circulation by using coupled model experiments with modified albedo parameters. When only studying model experiments with prescribed year-round sea 505 ice loss, more dynamical based analyses (e.g. Jaiser et al., 2016) have to be conducted in order to assess the role of stratospheric pathways and autumn sea ice loss. This was however not the focus of the present study.

Furthermore, we use an atmosphere-only model that does not allow for a representation of atmosphere–ocean feedbacks. In this respect, previous studies stressed the importance of an interactive ocean model (Screen et al., 2018). This may allow for representing additional oceanic pathways such as altered ocean currents and have shown to amplify circulation responses to 510 Arctic sea ice loss. However, in contradiction to this hypothesis a recent study by England et al. (2022) shows that different approaches that impose sea ice perturbations in a coupled model setup add artificial heat to the Arctic region. This causes a spurious warming signal that is added to the warming expected from sea ice loss alone, and therefore finally results in an overestimation of the climate response to sea ice retreat in coupled model setups.

The atmospheric response to sea ice loss also depends on the exact prescribed patterns of sea ice and SST boundary forcing 515 (Screen, 2017a; McKenna et al., 2018) and the used model. Crasemann et al. (2017) for instance studied sea ice sensitivity simulations conducted with the general circulation model for Earth Simulator (AFES, Nakamura et al. (2015)). Compared to the experiments used in our study their simulation data consisted of two perpetual runs over 60 years, however forced under sea ice conditions averaged over the early 80th and the early 2000th respectively. Additionally, their SST background states were set to the early 80th. With respect to circulation regime changes they detected an increase of the Scandinavian blocking pattern 520 under low sea ice conditions already in December, as well as a more frequent occurrence of the NAO- pattern in February and March.

The five circulation regimes that were used throughout the study only provide coarse categorizations of the atmospheric flow and contain a variety of more specific synoptic patterns. In the case of European winter temperature extremes we discussed





that some of these few large-scale variability patterns might be suitable in order to describe the typical atmospheric circulation
during such extremes, or at least contain most of the relevant synoptic patterns. The atmospheric situations during e.g. spatially
confined precipitation extremes, as well as summer heatwaves that typically co-occur with an atmospheric ridge may be to
unique and uncommon in order to be examined and allocated to a certain large-scale circulation regime. An analogue approach
might be more suitable for such extremes.

The framework of conditional extreme event attribution employed in this study provides only one unique way to decompose
atmospheric responses. The individual decompositions assume that the occurrence of a certain extreme can be completely
associated with the presence and changes of a certain circulation regime. Studies by Vautard et al. (2016) or Cassano et al.
(2007) proposed for instance an approach where the individual contribution terms related to specific regimes add up to the
overall response. However, Vautard et al. (2016) also showed very limited suitability of this methods when working with a very
small number of circulation regimes.

It should be noted that within this study we only considered changes in the occurrence probability of extremes defined
by a fixed threshold temperature in a present day simulation. Similarly, changes in circulation regimes have also only been
considered in terms of frequency changes. When aiming to draw conclusions about changes in the intensity and severity of
extremes, other factors have to be taken account such as the actual strength of advection processes. Therefore, it might be
helpful to distinguish between days that strongly (weakly) project onto a relevant pattern (e.g. NAO- for cold extremes) and
are therefore connected to stronger (weaker) advective processes. This may provide refinement possibilities of the approach
employed here for upcoming studies.

Nevertheless, the present study provides a complementary and useful perspective on the question how future Arctic sea ice
retreat can impact atmospheric large-scale dynamics, as well as to what extent European temperature extremes are affected by
future Arctic sea ice loss and how these changes can be separated into by dynamically and thermodynamically contributing
factors.



## Appendix A

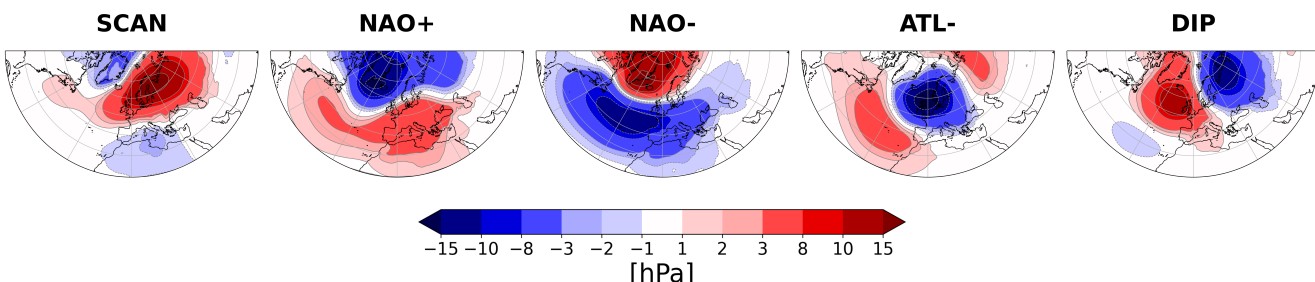

**Figure A1.** Five circulation regimes over the Euro-Atlantic domain computed from daily ERA5 sea level pressure anomaly data (1979–2018) for extended winter season (December, January, February, March).

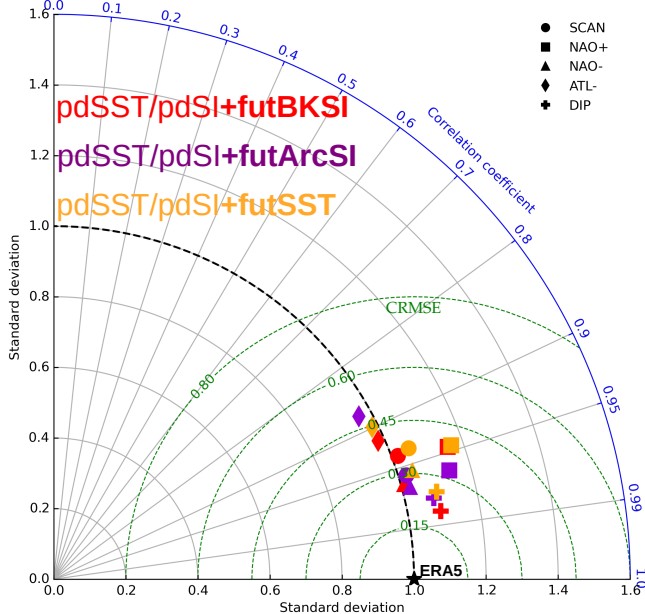

**Figure A2.** Taylor diagram (Taylor, 2001) that summarizes different statistics in order to compare computed model patterns with regime patterns obtained from ERA5. Different symbols indicate different regimes and different colors stand for different combinations of model simulations for which regimes are computed in this study. The black star symbolically indicates the ERA5 reference pattern. The concentric quadrants centered around the origin show the pattern standard deviation of the different model patterns relative to the standard deviation of the ERA5 reference patterns. The blue polar axis depicts the pattern correlation coefficient between the respective model patterns and the reanalysis pattern. The green concentric semicircles centered around the black reference point indicate the centered root mean square error (CRMSE) when comparing model and reanalysis patterns. Thus, model symbols close the reference star mean high resemblance between model and reanalysis pattern.



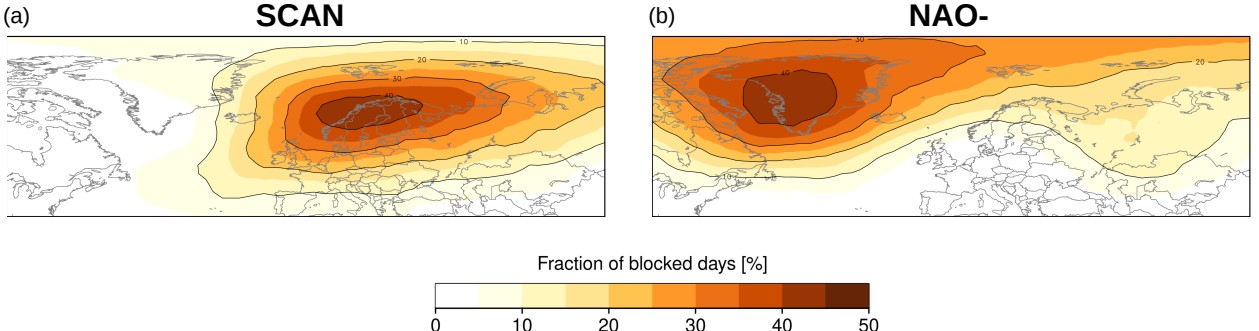

**Figure A3.** Mean DJFM relative blocking frequency (fraction of blocked days) at the same time a SCAN or negative NAO regime is present in ECHAM6 PAMIP pdSST/pdSI simulation. Blocking frequency is calculated based on a slightly modified version of the two-dimensional blocking index from Scherrer et al. (2006). Based on gradients in the daily 500 hPa geopotential height field and areas of positive gph anomalies associated with the blocking detection described in Schuster et al. (2019), daily blocked grid points are identified.

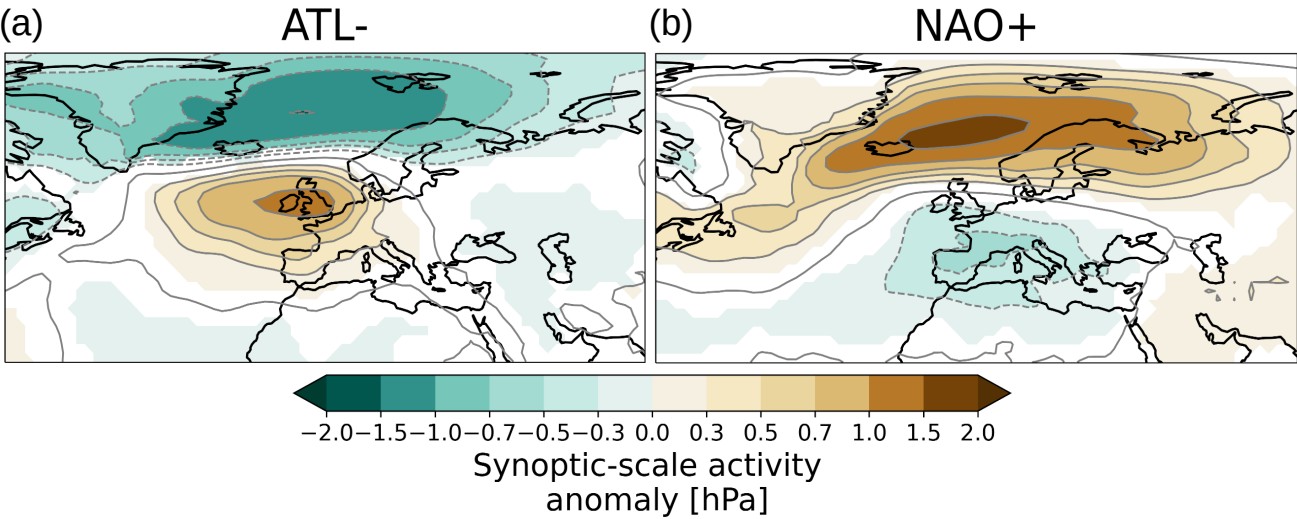

**Figure A4.** Synoptic-scale activity anomalies (DJFM) for the ATL- and NAO+ regimes computed from PAMIP pdSST/pdSI model data. Synoptic-scale activity is computed here as the 2–6 day bandpass filtered standard deviation of slp data (Blackmon, 1976). It provides a measure for baroclinic activity and characterizes stormtrack locations. Only anomalies that significantly differ from zero according are shown in colors.




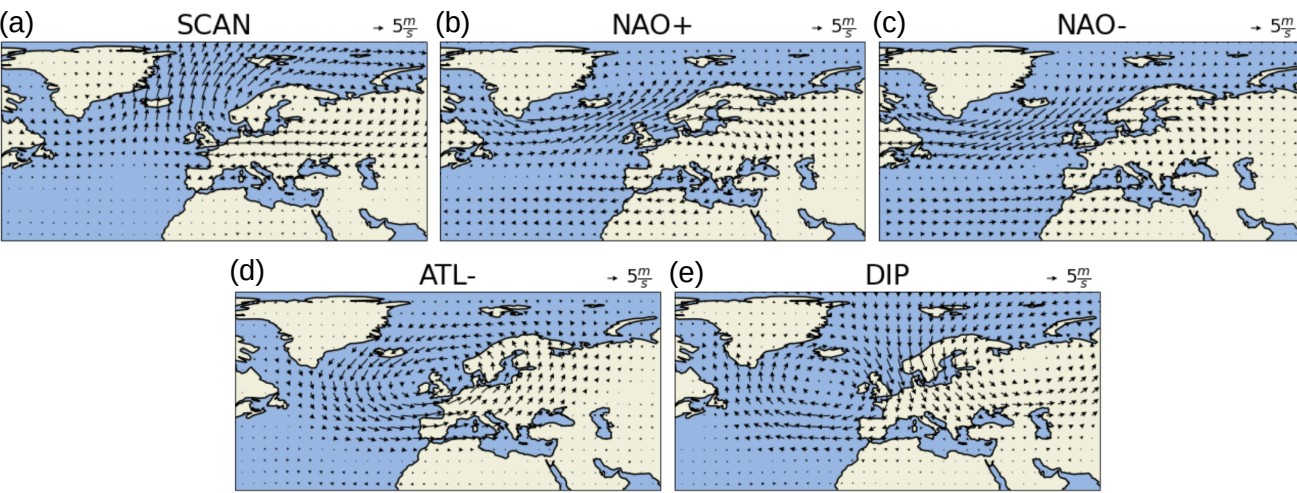

**Figure A5.** Wind anomalies at 700 hPa (DJFM) for the circulation regimes computed from PAMIP pdSST/pdSI model data.

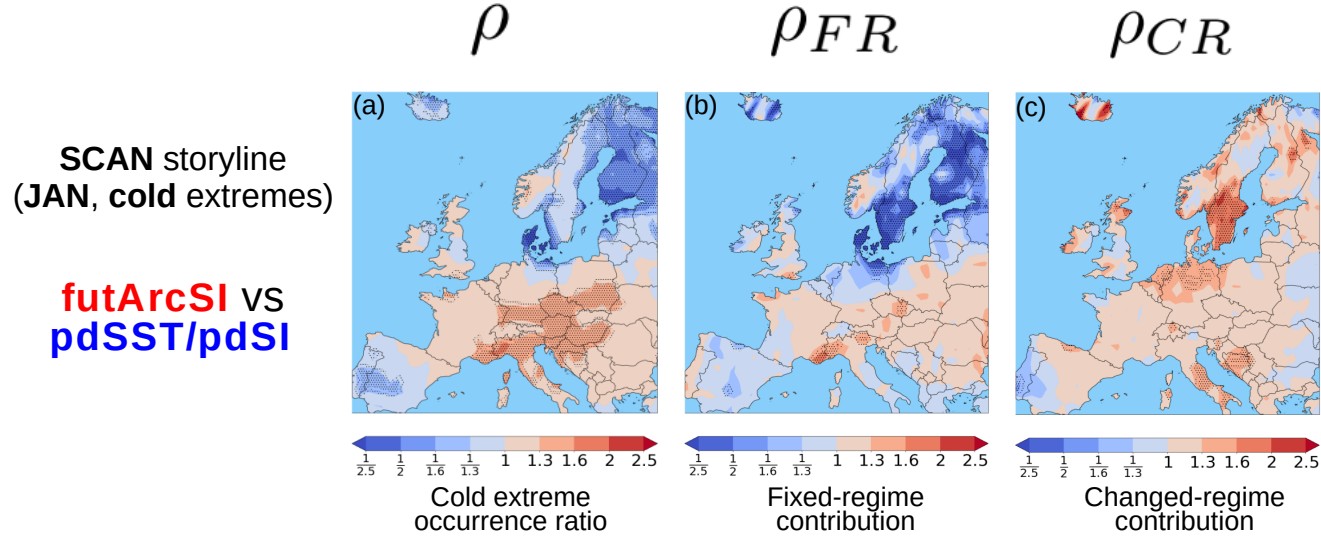

**Figure A6.** Same as in Figure 6 but comparing the futArcSI and pdSST/pdSI simulations, and only for January cold extremes along a SCAN regime storyline. Occurrence ratio of SCAN regime occurrence in January is given as $\rho_{circ} = 1.17$. Thus, the SCAN occurs more frequent in the futArcSI simulation (see also Figure 2a).



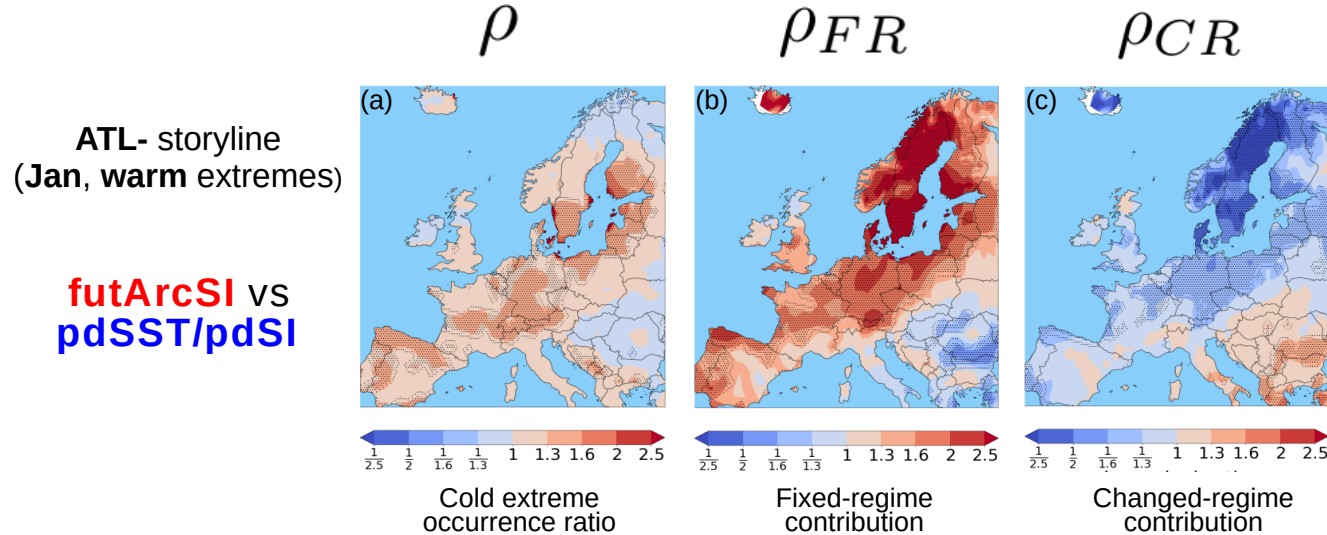

**Figure A7.** Same as in Figure 6 but comparing the futArcSI and pdSST/pdSI simulations, and for January warm extremes along a ATL-regime storyline. Occurrence ratio of ATL- regime occurrence in January is given as $\rho_{circ} = 0.79$. Thus, the ATL- regime occurs less frequent in the futArcSI simulation (see also Figure 2d)



*Data availability.* ERA5 data have been stored and accessed via the Deutsches Klimarechenzentrum (DKRZ) in Hamburg. The model simulations were conducted, originally stored and accessed at the DKRZ as well, but are also available for instance at the system of the Earth System Grid Federation (ESGF)

550 *Author contributions.* DH and JR developed the original idea for the paper. JR conducted the analysis and wrote the original draft. AR did the blocking frequency plots. AR, HR, UU and DH supervised and contributed to the interpretation of the results and provided feedback for the manuscript. TD did the ECHAM6 model simulation and also provided feedback for the manuscript.

*Competing interests.* The authors declare that they have no conflict of interest.

*Acknowledgements.* Johannes Riebold, Dörthe Handorf, Andy Richling, Henning Rust, Uwe Ulbrich gratefully acknowledge the support by
555 the ClimXtreme project, subproject ArcClimEx, funded by the German Ministry of Research and Education (grant 01LP1901D (JR, DH) and 01LP1901C (AR, HR, UU). Dörthe Handorf was partly supported by the German Research Foundation (DFG, Deutsche Forschungsgemeinschaft) Transregional Collaborative Research Center SFB/TRR 172 "Arctic Amplification: Climate Relevant Atmospheric and Surface Processes, and Feedback Mechanisms (AC)3" (Project-ID 268020496) and by the European Union's Horizon 2020 research and innovation framework programme under Grant agreement no. 101003590 (PolarRES). For his PAMIP related work, Tido Semmler gratefully
560 acknowledges the support by the EU H2020 APPLICATE project (GA727862). Finally, the authors also want to acknowledge the Deutsches Klimarechenzentrum (DKRZ) in Hamburg for providing the general technical infrastructure for the analysis.

*Financial support.* This research has been supported by the German Ministry of Research and Education BMBF(ClimXtreme, grants 01LP1901D and 01LP1901C ), the German Research Foundation DFG (Transregional Collaborative Research Center SFB/TRR 172 "Arctic Amplification: Climate Relevant Atmospheric and Surface Processes, and Feedback Mechanisms
565 (AC)3", Project-ID 268020496), the European Union's Horizon 2020 research and innovation framework programme under Grant agreement no. 101003590 (PolarRES), as well as the EU H2020 APPLICATE project (GA727862).



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
