# Peer review of "On the linkage between future Arctic sea ice retreat, Euro-Atlantic circulation regimes and temperature extremes over Europe"

_EGUsphere, 2022_

## Author Comment (AC2)

**Supplementary material**

**Table S1:** List of PAMIP models that were used for the analysis in Figs. S1–S3. Listed is also the available ensemble size for each experimental setup (in number of years)

| Institute | Model | pdSST/pdSIC | futArcSIC | futBKSIC |
|---|---|---|---|---|
| Max Planck Institute for Meteorology, Hamburg | ECHAM6 | 100 | 100 | 100 |
| National Center for Atmospheric Research Canadian | CESM2 | 200 | 200 | - |
| Norwegian Meteorological Institute | NorESM2-LM | 200 | 200 | - |
| University of Tokyo/National Institute for Environmental Studies/Japan Agency for Marine-Earth | MIROC6 | 100 | 100 | 100 |
| Canadian Centre for Climate Modelling and Analysis | CanESM5 | 300 | 300 | - |
| US Department of Energy/University of California Irvine | E3SMv1 | 185 | 190 | 100 |
| Met Office UK | HadGEM3-GC31-MM | 300 | 300 | 200 |
| National Center of Atmospheric Research/ University of California Irvine | CESM1-WACCM-SC | 300 | 100 | 100 |
| Institute of Atmospheric Physics, Beijing | FGOALS-f3-L | 100 | 100 | - |
| Centre Européen de Recherche et de Formation Avancée en Calcul Scientifique US | CNRM-CM6-1 | 300 | 300 | - |
| Institute Pierre Simon Laplace University | IPSL-CM6A-LR | 200 | 200 | 100 |

[Figure]

**Figure S1:** Taylor diagrams that summarize different statistics in order to compare computed model patterns from different PAMIP models with regime patterns obtained from ERA5 (see e.g. Fig. A2). Different symbols/colors indicate different models. Regimes were computed by merging the pdSST/pdSIC with the futArcSIC data set. Individual Taylor diagrams compare individual model regimes with the respective ERA5 regimes. The Taylor diagram in the lower right ("all") compares the statistics averaged over all regimes.

[Figure]

**Figure S2:** Relative regime occurrence frequencies for different winter months as in Figs. 2a–e, but for different PAMIP models (ERA5 not shown). Compared are the pdSST/pdSIC reference simulation (blueish bars) and the futArcSIC (redish bars). Only the nine models that according to Fig. S1 are able to realistically reproduce most of the ERA5 regime pattern structures were considered here. Dark-colored bars indicate significant differences.

[Figure]

**Figure S3:** Relative regime occurrence frequencies for different winter months similar to Figs. 2f–j, but for different PAMIP models. Compared are the pdSST/pdSIC reference simulation (blueish bars) and the futBKSIC (redish bars). Only the five available models that according to Fig. S1 are able to realistically reproduce the regime pattern structure were considered here. Dark-colored bars indicate significant differences.

**Table S2**: Absolute number of regime days for each winter month that were used for the computation of the ERA5 regime frequencies in Fig. 2. Blue numbers indicate the number of regime days for high Arctic sea ice conditions, red numbers for low sea ice conditions.

|  | SCAN | NAO+ | NAO- | ATL- | DIP |
|---|---|---|---|---|---|
| December | 112/168 | 115/124 | 94/119 | 156/102 | 143/107 |
| January | 81/181 | 151/100 | 94/110 | 163/100 | 131/129 |
| February | 100/183 | 142/82 | 94/94 | 96/119 | 128/82 |
| March | 98/147 | 142/137 | 114/127 | 107/116 | 159/93. |

**Table S3**: Absolute number of regime days for each winter month that were used for the computation of the relative frequencies in Figs. 2a–e. Blue numbers indicate the day count for the (100 year-long) ECHAM6 pdSST/pdSIC experiment, red numbers for the (100 year-long) ECHAM6 futArcSIC experiment. Note, that the sum over all regimes for a specific month and an experiment sums up to 100 times the number of days within the respective month (e.g. 100*31 days =3100 days for December).

|  | SCAN | NAO+ | NAO- | ATL- | DIP |
|---|---|---|---|---|---|
| December | 676/699 | 576/521 | 540/552 | 657/702 | 651/626 |
| January | 585/682 | 610/569 | 522/596 | 770/611 | 613/642 |
| February | 534/572 | 536/594 | 682/455 | 484/660 | 564/519 |
| March | 693/619 | 494/671 | 612/566 | 674/654 | 627/590 |

**Table S4**: Same as Table S3 but for pdSST/pdSIC and futBKSIC. Blue numbers indicate the day count for the (100 year-long) ECHAM6 pdSST/pdSIC experiment, red numbers indicate the counts for the (100 year-long) ECHAM6 futBKSIC experiment that were used for the computation of the relative frequencies in Figs. 2f–j .

|  | SCAN | NAO+ | NAO- | ATL- | DIP |
|---|---|---|---|---|---|
| December | 706/ 666 | 546/ 521 | 564/ 567 | 662/ 654 | 622/ 692 |
| January | 567/ 703 | 585/ 537 | 534/ 605 | 792/ 634 | 622/ 621 |
| February | 498/ 612 | 598/ 482 | 646/ 524 | 505/ 652 | 553/ 530 |
| March | 700/ 618 | 518/ 542 | 585/ 619 | 679/ 666 | 618/ 655 |

[Figure]

**Figure S4:** Same analysis as in Fig. 3, but for ERA5 over the period 1979–2018. Thus, regime patterns computed from ERA5 were used for computing these plots. ERA5 T2max/T2min times series at each grid point were linearly detrended beforehand.

---

## Author Response (AR1)

**General response**

*We thank both anonymous reviewers for carefully reading our manuscript and providing very useful feedback that helps to improve the manuscript's quality.*

*Based on the reviewer's comments we revised the manuscript appropriately. As a major modification we repeated the analysis of regime frequency changes for other PAMIP models and incorporated the results into the storyline of the manuscript. In addition to the issues raised by the reviewers, we detected and did some few additional minor corrections (typos, clarifications of formulations etc.).*

*Below we provide a one-to-one response to all points raised by the reviewers. The reviewers' comments are in red and our replies are in blue. All line numbers refer to the revised manuscript.*

**Point-to-point response to Reviewer #1**

Review for "On the linkage between future Arctic sea ice retreat, Euro-Atlantic circulation regimes and temperature extremes over Europe" by Riebold et al.

Recommendation: minor revision

Summary

In this manuscript, the authors analyze the Polar Amplification Intercomparison Project experiments with ECHAM6, focusing on the response of five Euro-Atlantic winter circulation regimes to future Arctic sea ice reduction, and their linkage to cold/warm extremes. Some significant regime frequency changes have been identified such as more frequent occurrences of a Scandinavian blocking pattern in midwinter under reduced sea ice conditions. The authors also decomposed the sea ice induced winter extreme temperature frequency change into thermodynamical and dynamical changes. They compared the results with SST-induced results and found that Arctic sea ice loss-induced effect is of secondary relevance. Overall, I found this manuscript interesting and well fits the scope of Weather and Climate Dynamics. I also have some comments and hopefully they can help improve the manuscript. Minor revision is recommended.

We thank reviewer #1 for his/her overall positive feedback.

Major comments:

1. I am interested in the conditional extreme event attribution framework that the authors presented. According to the section 3.2, the total ratio can be decomposed into thermodynamical component and dynamical component within the framework of circulation regimes. This is somewhat similar to the previous dynamical adjustment work. However, I find it hard to interpret how the thermodynamical processes contribute to the increase in cold extremes (e.g., Fig. 6b Eastern Europe). I wonder if this implies some unexplained physical processes or simply noise. In other words, why is it still common to observe increased cold (decrease warm) extremes in Fig. 6, 7, 8, and A6, A7?

Such increases in cold extremes could be related to dynamical changes in the atmospheric circulation that are captured by the fixed-regime contribution. The fixed-regime term compares extreme occurrences under fixed atmospheric dynamics in terms of circulation regimes. Although we assessed the similarity of regime structures for different experiments in Fig. A2, as mentioned in line 235 Individual daily flows that are allocated to some regime can still differ between different experiments. Hence, it can not be ruled out that daily flows in the future experiment that are allocated to a respective regime change in a way that they more (less) frequently promote the occurrence of cold (warm) extremes over certain regions.

For this reason, the authors intentionally renamed the "thermodynamical" and "dynamical" contributions (as termed in the original methodology paper by Yiou et. al. (2017))  into "fixed-regime" and "changed-regime" contributions. This was done in order to avoid  too direct associations with purely thermodynamical and dynamical processes when interpreting both terms.
In this respect, we now also use the terms "Fixed"-and "Changed-Regimes" more consistently throughout the revised manuscript.

In  line 431 we added a discussion of this issue for Fig. 6b: *"rho_FR compares the extreme occurrence probability during SCAN days. Hence, it cannot be ruled out that the individual daily flow patterns allocated to the SCAN regime change in a way that they more frequently promote the occurrence of southwestward cold air advection towards Eastern Europe, and thus, also the occurrence of cold extremes over this region."*

Also, maybe this is because I didn't fully understand section 3.2, I wonder why the  is different from the regime pattern. For example, Fig. 3a indicates that SCAN tends to causes increased cold extreme occurrence in central Europe and decreased occurrence in UK, Ireland, Iceland and Northern Europe, while Fig. 6c showed that in response to Arctic sea ice loss, associated with increased SCAN frequency (Fig. 2a) there is increased frequency of cold extremes in central Europe, Ireland, UK and Iceland. I would encourage the authors to elaborate on these issues for readers to better understand this approach and results.

This issue is related to the underlying assumption of the decomposition method that the presence of the reference regime is necessary for an extreme to occur.
Consequently, when bearing this assumption in mind, it is only reasonable to interpret the contributions over regions where according to Fig. 3 the extreme event occurs anomalously often during the occurrence of the respective regime. On the one hand, Fig 3a shows anomalously more frequent occurrences of cold extremes over Central Europe during

SCAN; hence, the underlying assumption of the decomposition is well justified and the results over central Europe in Fig. 6c can be interpreted as mentioned by the reviewer. On the other hand, for cold extremes over the UK, Iceland etc. the underlying assumption is less justified, as cold extremes over these regions barely occur during SCAN days. Consequently, an interpretation of the contribution terms over these regions in Fig. 6c is not very sound.

In order to avoid such potential misconceptions regarding the interpretation of the results, we updated the decomposition Plots (Figs. 6–9 and A6 and A7) in a way that both contribution terms are only shown over European regions for which Fig. 3 shows statistically significant more frequent (greater than one) extreme occurrence ratios during the considered reference regime.
For further clarification, we substituted the following text in line 419: *" The employed decomposition method assumes that the presence of the respective reference regime C_ref is necessary for an extreme to occur; hence, rho_CR and rho_FR are only plotted over regions where Fig. 3 indicates statistically significant more frequent extreme occurrences during C_ref."*

2. The authors noticed that "reductions in cold extreme occurrences are not necessarily accompanied by less frequent occurrences of warm extremes, and vice versa" (line 475), and demonstrated that "Such asymmetric responses in the tails of the temperature distributions can not be explained by simple thermodynamical arguments and are certainly a result of other contributing factors such as changes in the dynamical situation leading to a certain extreme." (line 365). This reminds me an earlier paper by Screen (2014), who found that Arctic sea ice loss decrease midlatitude temperature variability because northerly winds and associated cold days are warming more rapidly than southerly winds and warm days. In this context, I wonder if asymmetric response is still likely caused by thermodynamical argument.

Thanks for the remark. For sure, it is also plausible that, on the one hand, an overall stronger warming of northerly winds compared to southerly winds (as argued by Screen (2014)) partly contributes to this asymmetric response in extreme occurrences. On the other hand, more frequent occurrences of for instance northward compared to southward advective situations may dynamically also contribute to such an asymmetric response .

Line 400 now states: *"Such asymmetric responses in the tails of the temperature distributions could be thermodynamically explained by a stronger warming of northerly polar winds compared to southerly winds as argued by Screen (2014). Nevertheless, such responses could also be a result of other contributing factors, such as changes in the occurrence frequencies of atmospheric flows leading to certain extremes."*

Specific comments:

1. Sampling issue: while 100 members are the recommended sample size for polar amplification-model intercomparison project, recent studies have found that atmospheric response to Arctic sea ice loss still subjects to large uncertainty even with 100 members (e.g., Peings et al. 2021; Streffing et al. 2021; Sun et al. 2022). I don't think the authors need

to rerun another 100 members, but just feel that this is one caveat that should be kept in mind.

Thanks for the remark. We added a paragraph in the concluding remarks (line 541):

*"First, the presented analysis for ECHAM6 was conducted based on 100 ensemble members of one-year-long time slice simulations for each respective experimental setup. In this respect, recent studies by Streffing et. al. (2021), Peings et. al. (2021) and Sun et. al. (2022) suggested that 100 ensemble members may not be enough in order to isolate the forced mean response from internal atmospheric variability in PAMIP sea ice sensitivity experiments."*

2. Line 170: this is very minor since the authors mentioned that it does not matter whether the individual or merged climatology is used. But I do wonder if there is a reason for the authors to prefer using merged climatology. My understanding is that the climatology between pdSI and futArcSI might be very similar in the midlatitude, but might not in the Arctic. Therefore, using individual climatology appears to be better unless they have other considerations.

We think that there is no definite "better" or "worse" when contrasting an annual cycle obtained by removing individual climatologies with an annual cycle computed by removing merged climatologies. Our basic philosophy was to retain as much information of the original data as possible when applying the cluster analysis and comparing the regime occurrence frequencies (unless the background pattern strongly projects onto one of the regime structures and extremely biases the analysis of regime frequency changes, as it was the case for futSST).

Editorial comments:

Line 115: sea surface temperature (SST)

We changed "SST" to "sea surface temperature (SST)"

Line 120: readers will benefit if the authors can provide a very brief description of ECHAM6

Thanks for the remark. In line 132 we now provide a brief description of ECHAM6:

*"ECHAM6 is the latest release of the atmospheric general circulation model ECHAM that was developed at the Max-Planck-Institute for Meteorology (MPI) in Hamburg (Stevens et. al. 2013). The ECHAM6 setup used for the PAMIP experiments operates on 95 vertical layers up to 0.01 hPa, and with a spectral T127 horizontal resolution (resulting in a zonal resolution of 100 km in the tropics and for instance 25 km at 75°N)."*

Line 160: Is PCA principal component analysis?

Yes. We changed "PCA" to "principal component analysis".

Throughout the manuscript (e.g., lines 130, 155, 165, 200) the authors use pdSI, futArcSI and futBKSI, I suggest to use "SIC" so as to be consistent with the PAMIP convention.

Thanks for the suggestion, we now use pdSIC, futArcSIC and futBKSIC in the revised manuscript.

Line 175: It is hard for me to understand why global SST warming is causing negative phase of the NAO. Shouldn't it be positive NAO (e.g., Fig. 8 of Blackport and Kushner 2017; Fig. 8 of Sun et al. 2018)?

Thank you very much for the remark, this was actually just a typo. The change in the slp background state projects onto the positive NAO. We corrected this.

**References:**

Stevens, Bjorn, et al. "Atmospheric component of the MPI‑M Earth system model: ECHAM6." *Journal of Advances in Modeling Earth Systems* 5.2 (2013): 146-172.

Screen, James A. "Arctic amplification decreases temperature variance in northern mid-to high-latitudes." *Nature Climate Change* 4.7 (2014): 577-582.

Streffing, J., Semmler, T., Zampieri, L., & Jung, T. (2021). Response of Northern Hemisphere weather and climate to Arctic sea ice decline: Resolution independence in Polar Amplification Model Intercomparison Project (PAMIP) simulations, Journal of Climate.

Peings, Y., Labe, Z. M., & Magnusdottir, G. (2021). Are 100 ensemble members enough to capture the remote atmospheric response to+ 2°C Arctic sea ice loss?. Journal of Climate, 34(10), 3751-3769.

Yiou, Pascal, et al. "A statistical framework for conditional extreme event attribution." *Advances in Statistical Climatology, Meteorology and Oceanography* 3.1 (2017): 17-31.

**Point-to-point response to Reviewer #2**

General comments:

This study examines how projected Arctic sea ice decline might affect the large-scale atmospheric circulation over the Euro-Atlantic region in terms of frequency of occurrence of weather regimes and temperature extremes. It is based on the analysis of sensitivity experiments conducted with the ECHAM6 atmospheric model within the framework of CMIP6 PAMIP coordinated experiments. Several sets of experiments are analyzed: present-day simulations (pdSI/pdSST) and idealized simulations in which Arctic sea ice is reduced either over the whole Pan-Arctic region (futArcSI) or only in the Barents/Kara Sea region (futBKSI). Each experiment consists of 100 members of 1 year. In order to assess the role of future Arctic sea ice reduction on large-scale atmospheric circulation, five weather regimes over the Euro-Atlantic regions are computed and their relationship with cold and warm temperature extremes is examined. The authors show that the frequency occurrence of three weather regimes, SCAN, NAO+ and ATL is affected by Arctic sea ice reduction. The change in the frequency of occurrence in the model experiments is compared to observations using ERA5. The authors compute the regime occurrence frequency in ERA5 for lower than averaged and above average Arctic sea ice conditions and compare these two situations with the present day and future simulations. Only the Scandinavian Blocking and the Atlantic Ridge regime show some significant results that are comparable to observations and in general only for one month among the 4 winter months. Hence the signal appears to be quite weak and only detectable for specific months. The comparison between futArcSI and futBKSI indicates that most of the frequency changes can be explained by the regional contribution of the Barents/Kara Sea sea ice reduction. The authors then apply a storyline approach using the conditional extreme event attribution framework described in Yiou et al. (2017) to identify the respective contribution of dynamical and non-dynamical changes in the modeled response of extremes to see ice reduction. They show that European cold extremes during winter can be mainly attributed to changes in the occurrence of the Scandinavian blocking as well as to a non-dynamical thermodynamical component. The authors also compare the sea-ice induced atmospheric changes to global increase of SST to evaluate the importance of Arctic sea ice decline in future climate changes. This comparison suggests that sea-ice decline is of secondary importance compared to future SST change.

The topic of the paper is important because the role of Arctic sea ice loss on midlatitude climate is highly debated and deserves attention. The analysis conducted in this study are very thorough with a comprehensive description of the mechanisms that might be at play in the atmospheric response to sea ice reduction. The paper is well written, I really enjoyed reading it.

We thank reviewer #2 for his/her positive feedback.

Some of the figures could have been clearer in particular the statistical significance that is often difficult to see in most figures.

Thanks for the feedback, we improved/updated Figures 1–9 and A6, A7 in the revised manuscript by incorporating the reviewer's suggestions.

My main concern is the fact that the paper is based only on the analysis of one model experiments. While I can understand the value of analyzing experiments from a single model when it is the first time a protocol is used, the experiments analyzed in this paper have been conducted by many climate centers as part of the coordinated multi-model PAMIP within CMIP6 and hence not taking advantage of this unique database is to my opinion a strong weakness of this study. This is even more important that several studies have shown that 100 members were not enough to show a robust response to Arctic sea ice decline (e.g. Peings et al. 2021) and that models may underestimate the atmospheric response to sea ice loss (Smith et al. 2022). The authors themselves state in their conclusion that "the sign and significance of the signals highly depend on the respective month". Having several models and more members would likely have increased the signal to noise ratio and could have allowed to see a more robust response in terms of changes in weather regimes frequency occurrences and the associated temperature extremes. Hence, I strongly recommend extending the analysis conducted in this study to more models before allowing the publication of this paper.

We retrieved all PAMIP model data for which daily slp data were available to us from the ESGF in January 2023: 10 additional models for the futArcSIC and pdSST/pdSIC experiments, as well as five additional models for the futBKSIC experiment (see Table S1).

We repeated the analysis of regime frequency changes in Sec. 4.1 for these PAMIP models. We revised and restructured Sec. 4.1 in a way that we now discuss the regime frequency changes found in other PAMIP models as well. In order to structure Sec. 4.1 more clearly, we divided Sec. 4.1 into subsection 4.1.1 *"Regime structures"* and subsection 4.1.2 *"Regime frequency changes induced by future Arctic sea ice retreat".*
In Sec. 4.1.1 we initially present and discuss the ECHAM6 regimes structures (lines 272-292), but also discuss in lines 293-299 the ability of other PAMIP models to reproduce the ERA5 regime structures (see Taylor diagrams in Fig S.2). Afterwards, in Sec. 4.1.2 (lines 300-325) we now initially discuss regime frequency changes for nine PAMIP models that are able to realistically reproduce the ERA5 regime structures (Fig S2 and S3), and subsequently constrain the frequency changes in ECHAM6 by recent ERA5 tendencies for the later use in Sec. 4.4 (lines 326-339).

In order to demonstrate how such previously detected regime frequency changes can be employed to decompose changes in extremes into Fixed- and Changed-Regime contributions, the focus of the analyses in Secs. 4.2–4.4 remains on the ECHAM6 model. In this respect, we now explicitly stress different aspects that motivate the choice of the ECHAM6 model for the remaining part of the manuscript: first, in line 328 the choice of ECHAM6 is motivated by the fact that it is among the best models that realistically reproduce the ERA5 regime structures. Secondly, as outlined in line 345 the identified links in ECHAM6 between regime and temperature extreme occurrences over different European regions

(see Fig. 3) show pronounced similarities with links identified in the ERA5 reanalysis (see Fig. S4).
In the conclusion, we suggest that the possibility to extend the employed decomposition methodology for a feasible implementation into a multimodel analysis is left open  as a potential prospect for future studies  (line 545).

More detailed comments:

l.1-24: The abstract is quite long and dense. I suggest reducing it to better emphasize the novelty of the work described in the paper.

Thanks for the feedback, we shortened and revised the abstract in the revised manuscript.

-l.6, l.43, and at many other places in the manuscript, the term Barents/Kara Sea is written Barents/Karasea. I suggest writing Kara Sea with two words.

Thanks for the remark, we changed  this throughout the manuscript.

l.40: A reference to Smith et al. (2022) should be added here as they analyze the wave activity response to Arctic sea ice reduction in about 16 models and provide an emergent constrain based on eddy feedback.

Thanks for the remark, we included the paper.

L45: I suggest adding here a reference to Blackport and Screen (2020) who also addressed extensively the lack of consensus about sea-ice induced atmospheric linkages.

Thanks for the suggestion, we added the reference.

-l.60 "effecting" should be replaced by "affecting"

Thanks for the remark, we corrected this.

-l.66: "effected" should be replaced by "affected"

Thanks for the remark, we corrected this.

-l.75 " "Climate model simulations typically suffer low signal-to-noise ratio" . It would be relevant to add here two references: Smith et al. (2022) and Scaife et al. (2018).

Thanks for the suggestion, we added both references.

l.75-77: This sentence would strongly support the use of more than one model to address issues like the one investigated in this paper.

As discussed in the major comment above we repeated the analysis of regime frequency changes for other PAMIP models.

-l.80-90: I suggest adding here a reference to the work of Gervais et al. (2016) and compared the results of this study to those found in this paper.

Gervais et.al. (2016) hypothesize "that changes in the surface forcing by sea ice and SSTs could lead to changes in boundary conditions that alter the frequency of occurrence of (circulation) patterns". Therefore, we added the reference in line 54 as an example how local forcings, such as local Sea ice changes, can impact the occurrence probability of weather regimes.

-l.108 "analysis steps" is repeated twice.

Thanks for the remark, we corrected this.

-l.121: I don't understand why the author refer to this model set up as high resolution as T127 corresponds to about 1º ? Please clarify or provide the resolution in km or degree.

For the PAMIP experiments ECHAM6 operated in what is technically called a "HR" configuration, where "HR" stands for high resolution setup. We dropped this term, and now provide a short explanation of the spatial resolution in km in Sec. 2.

-l.123: Shouldn't we say "aims at" instead of "aims on" ?

Thanks for the remark, we corrected this.

-l.126-127: I suggest adding here that this is exactly what is recommended by the PAMIP protocol of Smith et al. (2019).

Thanks for the suggestion, we incorporated the reference.

-l.157: Can you explain a bit more why it is chosen to merge data from the two experiments to apply the cluster analysis, instead of doing it separately for the present day and future experiments?

The motivation behind merging both data sets was to obtain one common set of circulation regimes for both experiments, which could be used throughout the upcoming analysis steps. This procedure can be justified when we assume that the different forcings do not significantly impact the regime structure (see also Figure A2).

-l.164: "for 1000 times" should be replaced by " 1000 times"

Thanks, we corrected this.

-l.204: Pr is not explicitly defined. Also, I find the notation here with > et < on the same line slightly confusing. It would be easier to follow if warm and cold extreme events definitions were presented separately.

Thanks for the remark, we briefly defined Pr in the text and now better distinguish between warm and cold extremes. Therefore, we now present the extreme definition in (1) for cold and warm separately, specify the temperatures T (Tmin/Tmax) and threshold temperatures

for cold and warm extremes more precisely in Eqs. 1–5, and exemplary explain the decomposition procedure for cold extremes only.

-l.221: "terms" should be "term"

Thanks, we corrected this.

-Figure 1 caption: The acronyms of each circulation regime shown at the top of the panels could be explained in the caption.

Thanks for the suggestion, we included a brief explanation of the acronyms in the caption.

-l.257: I suggest adding "Arctic" in the title before "sea ice retreat"

Thanks, we included this suggestion.

-l.276: It would be useful to add the spatial correlation with ERA5 as a number over each panel of Figure 1.

We now display the spatial correlations with ERA5 in the plots.

-Figure 2 could be improved. The labels on the y-axis are difficult to read and could be made larger and/or in bold. The triangles showing ERA5 results are difficult to see when intercepting the vertical bar (red triangle on a red bar, blue on blue). The caption is also ambiguous in some places. The word "Transparent" for the non-significant colored bar is not easy to understand. I suggest using instead "Light" bars to contrast with the "darker" bars.

Thanks for the feedback, we incorporated the suggestions and improved Fig. 2 and its caption.

Further, the choice of a 50% threshold to classify ERA5 low and high sea ice conditions does not seem comparable to the present day and future experiments. The authors should justify this choice and use a threshold that is more relevant for the comparison with the model experiments.

The intention of the ERA5 analysis was to support the plausibility of the detected ECHAM6 regime frequency changes with links between sea ice and regimes that can be inferred from "real-world" data over recent decades.
In order to better isolate the impact of recent sea ice variations we considered linearly detrended ERA5 sea ice anomalies for this analysis. Detrending the sea ice times series allows to account for recent changes in regime occurrence frequencies due to sea ice variations, that are not explained by the recent linear trend in time (which contains the effects of all other facets of recent global climate change). Hence, a direct comparison in terms of total Arctic sea ice area between the sea ice fields in the experiments and the defined ERA5 low/high ice conditions is not possible. Modifying the ERA5 threshold would consequently not make the ERA5 low and high sea ice conditions more "comparable" to the present day and future experiments in terms of total Arctic sea ice area. For this reason, this is why we also call it "ERA5 tendencies".

As however also mentioned below, we repeated the ERA5 analysis for a modified threshold definition (25/75%) in order to assess the sensitivity of the ERA results to the threshold choice.

It would also be useful to add above each bar or in a separate table how many days are used for each regime and each month.

The supplementary Tables S2–S4 provide the total count of regime days for each month for ERA5 high and low sea ice phases, but also the absolute counts for the ECHAM6 futBKSIC, futArcSIC and pdSIC/pdSST experiments that were used for the computation of the bars in Fig. 2.

In this respect, we want to stress that the definition of ERA5 low and high states is actually based on sea ice area anomalies. This was not clearly mentioned in the original manuscript. We now clarified this in the text (line 332).

-In Figure 2, the authors choose to show the change of frequency occurrence for each month during the cold season. This results in only few situations where the fut experiments show significantly different results from the present-day experiment. The use of monthly means rather than seasonal means does not seem to be well justified. The use of seasonal means (DJFM) could have allowed to reduce the noise and potentially increase the signal to noise ratio. This is done in Figure 3 so I suggest replacing this figure with seasonal means or better justify the added value of using monthly means here.

We added the justification for the choice of monthly values for the occurrence frequency of the regimes in line 306:

*"We decided to analyze the regime occurrence for each winter month separately as proposed pathways underlying Arctic-midlatitude teleconnections are often characterized by their evolution over the autumn-winter season (e.g. Kretschmer et al., 2016, Siew et al., 2020)."*

As an example, the often discussed stratospheric pathway develops from autumn to late winter, whereby reduced sea ice in the Barents and Kara Seas favors the occurrence of Ural blocking in December, which induces enhanced upward wave activity in December and January, wave breaking in the stratosphere, leading to a weakening of the polar vortex in January/February and a transition of the NAO to its negative phase in February/March.

Further, as described in my general comment, I believe that at least this figure, if not all the figures shown in the paper, should be repeated using all the PAMIP model experiments that have provided daily data to check the robustness of the results and get more significant results.

As requested by the reviewer we repeated the analysis of regime frequency changes (Fig. 2) for all PAMIP models that are currently available to us (see also major comment above).

l. 303-304: The authors describe here a similar feature in ERA5 and in the models for the frequency of occurrence of NAO+ pattern. It would be informative to add whether these results are sensitive to the choice of the 50% threshold in observations. In other words, if a

70-80% threshold is used to defined high sea ice conditions in ERA5, would the similarity with model simulations for NAO+ in February and ATL- in January still hold?

We repeated the ERA5 regime frequency analysis with a modified threshold definition (lower 25 % of linearly detrended monthly averaged Arctic sea ice area anomalies for low sea ice conditions, upper 75% represent high sea ice conditions). It shows that also for this modified threshold the ERA5 occurrence frequency of NAO+ (ATL-) in February (January) is still significantly decreased under recent low sea ice conditions.

-Figure 3: the dots showing statistical significance are hardly visible and should be bigger or darker. Same remark for the hatching in Fig 4 and 5 and the stippling in Fig6, 7 and 8.

Thanks for the feedback, we updated the respective Figures 3–9, A6, A7 in a way that makes the statistical significance more visible.

-l.315: the use of "observed" does not seem to be appropriate. I suggest using instead "reported"

Thanks, we included this suggestion.

-l.328: "descend" should be replaced by "descent"

Thanks, we corrected this.

-l.341: This sentence does not seem to be written in good English. Please check and revise it if needed.

Thanks for the remark, we changed the sentence in line 376 to "The upcoming Section investigates the overall changes in occurrence frequencies of continental northern hemispheric winter temperature extremes, which can be expected under future Arctic sea ice loss in ECHAM6."

-Figure 4 and 5 and l.355 to 360: here again as in the description of Fig 2, it would have been interesting to comment which result remains robust when repeating the analysis using seasonal means rather than separately for each month.

As explained above we decided to investigate regime frequency changes for each month separately. As extremes are usually dynamically driven by atmospheric flow patterns (see also the storyline assumption for the decompositions in Sec. 4.4), we argue that it is reasonable to investigate extreme occurrence frequency changes on a monthly level as well.

-Figure 6: I do not see any stippling on this figure so it is difficult to assess which region shows significant changes in cold extremes. Please make the stippling more visible if there are any on this figure.

Thanks for the feedback. We replaced the stippling by hatchings and increased the overall resolution of the figure. Furthermore, in response to the feedback by reviewer #1 we only show the decomposition terms over European regions for which Fig. 3 shows statistically significant more frequent (greater than one) extreme occurrence ratios during the

considered reference regime. All other areas are masked in order to avoid misconceptions about the interpretation of the decomposition.

-l.431 and Fig.9: Results are shown here for NAO- regime. The authors say that similar results are found for other regimes. It would be good to explain why SST leads to similar pictures.

For clarification, after the description of Fig. 9 we now state in line 476:

*"Figure 9 illustrates the decomposition of changes in extreme occurrences only for a NAO-storyline, but results for other storylines reveal the same qualitative picture: the thermodynamical impact of globally increased SSTs dominates dynamical impacts related to regime frequency changes, regardless of the chosen reference regime."*

Is the frequency of occurrence of NAO- regime favored in FutBK ?

As illustrated in Fig. S3, favored frequency of the NAO- regime in futBKSIC can only be detected in one PAMIP model, three models actually indicate a significantly reduced NAO-frequency in futBKSIC.

-Figure 9: the caption is a bit confusing as it says that blue can refer to favored occurrences of cold extremes in pdSST/pdSI but reduced occurrences in futSST. I suggest clarifying the text by referring to what is shown on the panels

Thanks for the remark, for clarification we reformulated this part in the caption of Fig. 9:

*"Similar to Fig.6, but comparing the pdSST/pdSIC reference simulation and the futSST sensitivity simulation (blue indicates favored cold extreme occurrences in pdSST/pdSIC, red indicates favored cold extreme occurrences in futSST)."*

-l.509. This sentence sounds a bit ackward. I guess the words "and have shown" could be replaced by "that were shown"

Thanks for the remark, we corrected this.

-l.515: I suggest replacing "used model" by "model used"

Thanks, we included this suggestion.

-l.526: "to unique" should be replaced by "too unique"

Thanks, we corrected this.

**References:**

Smith, Doug M., et al. "Robust but weak winter atmospheric circulation response to future Arctic sea ice loss." *Nature communications* 13.1 (2022): 727.

Kretschmer, Marlene, et al. "Using causal effect networks to analyze different Arctic drivers of midlatitude winter circulation." *Journal of climate* 29.11 (2016): 4069-4081.

Siew, Peter Yu Feng, et al. "Intermittency of Arctic–mid-latitude teleconnections: stratospheric pathway between autumn sea ice and the winter North Atlantic Oscillation." *Weather and Climate Dynamics* 1.1 (2020): 261-275.

Scaife, A. A. & Smith, D. A signal-to-noise paradox in climate science. npj Clim. Atmos. Sci. 1, 28 (2018).

Gervais, Melissa & Atallah, Eyad & Gyakum, John & Tremblay, Bruno. (2016). Arctic Air Masses in a Warming World. Journal of Climate. 29. 160120095621001. 10.1175/JCLI-D-15-0499.1.

Blackport and Screen 2020, Insignificant effect of Arctic amplification on the amplitude of midlatitude atmospheric waves. Sci. Adv., 6, eaay2880, https://doi.org/10.1126/sciadv.aay2880.

---

## Author Response (AR2)

**General response**

*We thank the reviewer for carefully reading the revised manuscript and providing additional feedback in order to improve the manuscript's quality.*

*Below we provide a one-to-one response to all points raised by the reviewer. The reviewers' comments are in red and our replies are in blue. All line numbers refer to the newly revised manuscript.*

Second review of the paper by Riebold et al.

I appreciate the authors effort in addressing point by point the different comments I raised. Overall, I find the revised manuscript much clearer and improved. The authors have added an analysis of the different PAMIP model simulations as suggested in my first review. I find the multi-model results interesting but I regret that none of the figure showing multi-model results has been included in the main part of the manuscript. The multi-model PAMIP results are all in the supplementary material even though they are described in the main text. I suggest moving at least Fig. S2 to the main part of the manuscript. In addition, the different regime occurrence simulated by the different models (Fig. S2 and S3) could have been more extensively discussed before moving to the more in-depth analysis of the ECHAM6 results. This is important as it allows to stress which changes in regime occurrence frequency are robust and which ones are not.

Thanks for the remark, we now included Fig. S2 into the main text and describe and compare the regime occurrence frequencies simulated by the different models more extensively (l. 309-336).

Further, in addition to the 4 supplementary figures there are 7 figures in appendix and I do not really understand the value of keeping them separate. Are the figures in appendix supposed to be more essential to the understanding of the paper than those in supplementary? I suggest putting all the additional figures in one place (appendix or supplementary material) or to better justify the choice of keeping them separate.

Thanks for the remark, as also suggested by the co-editor we now moved the Appendix Figures, as well as the former Figure 2 into the Supplementaries.

The authors argue that choosing ECHAM6 for the subsequent analysis is justified by the fact that ECHAM6 is one of the models that compares best with ERA5 in terms of weather regimes patterns. I do not find this argument convincing as Figure S1 shows that other models than ECHAM6 (e.g. CNRM-CM6-1) show as good results and hence the analysis could have been done on a larger model subset. I understand that the authors do not want to conduct the dynamical adjustment analysis on the 9 PAMIP models but I think they should justify their choice differently. Maybe by saying that the temperature daily data were not available for all models (if it is true) or because the interpretation of 9 models is complicated

and having a first analysis on a single model can serve as a basis for a subsequent multi-model analysis. I do not really agree with this choice of using only one model but I think that the authors should at least provide a honest justification before accepting the paper for publication.

Thanks for the remark. As already indicated in the concluding remarks (l.560) and as correctly suggested by the reviewer, the interpretation of nine models is complicated and the single-model analysis of ECHAM6 can serve as a basis for a subsequent multi-model analysis. We now state this more explicitly in the main text in l. 339:

" We will focus on only one model as especially a comprehensive interpretation of the upcoming decompositions for all nine models is very challenging and beyond the scope of this study."

Detailed comments:

-l.10: add that this is true for at least one winter month

Done

-l.12: This is not consistent with Fig S2 which shows that for ECHAM6, MIROC, E3SM, CESM, the only significant changes are less NAO- days, not more. Please clarify.

The term "most models" was indeed not suitably used when summarizing the NAO response of the different models. We now mention this overall inconsistency of the NAO response between models in the abstract in the following way:

"Forced by future Arctic sea ice conditions, most models show more frequent occurrences of a Scandinavian blocking pattern in at least one winter month, whereas there is an overall disagreement between individual models on the sign of frequency changes of two regimes that respectively resemble the negative and positive phase of the North Atlantic Oscillation. "

-l.21-22: Fixed Regime is sometimes in the manuscript written with capital letters and sometimes not. Same for Changed Regime. Please make it consistent everywhere.

Done

-l.136: "over" should be replaced by "of"

Done

-l.138: "a certain number of members" should be replaced by "at least 100 members"

Done

-l.335: Please specify whether you count only the months where these changes are significant or whether you consider all months. Note that even the models that show an increase it is not true for all winter months. It could be worth stating it more clearly.

We now state this more clearly in the text (l.309)

"All nine models indicate  a significant increase of  SCAN occurrences  in futArcSIC in at least one winter month, while in contrast only the NorESM2-LM and CNRM-CM6-1 models show significantly decreased SCAN occurrences."

-l.354-355: It would be good to add here that ECHAM6 is not among those models, which means that ECHAM6 results are not consistent with the robust changes outlined in Smith et al. (2022).

In l. 327 we now explicitly  state that ECHAM6  indicates less frequent NAO- occurrences:

"In contrast, also decreased NAO- occurrences in at least one winter month can be detected in five models as well---including ECHAM6."

-l.363-364: I am not convinced by this argument as stated in my general comments.

See reply to major comment above. We now explicitly state as the main reason  of chosing one model that the interpretation of the implemented decomposition for all models would be challenging.

-l.372-373: It is quite confusing to describe the results of the reference simulation saying that it shows more frequent occurrence while in the rest of the section it is the sensitivity experiment changes that are described. I suggest keep the same method and describing everywhere the change in the sensitivity experiment (future sea ice) with respect to the reference experiment (present day sea ice). That would make the results description easier to follow.

Thanks for the remark, we now consistently  describe the changes with respect to the sensitivity experiment.

-l.374: Several studies in particular by Screen or Blackport have questioned the validity of using observations or reanalysis to detect the influence of sea ice on the atmosphere pointing out the difficulty to identify causal relationship. Hence, I suggest not insisting too much in the paper on the comparison between ERA5 and the model results (except when validating the weather regimes of course) especially given that the models themselves do not show consistent results for all the features that are described here.

As correctly pointed  out by the reviewer the complementary ERA5 regime analysis does not provide any insights into the actual causal sea ice- regime relationship, but  is meant  as some additional statistical evidence. We now explicitly state this in line 345:

"Such an ERA5 analysis does not prove any causal link between recent sea ice loss and circulation regimes, and does not isolate the effect of recent sea ice retreat. Nevertheless, we consider such ERA5 tendencies as additional statistical evidence, especially when deciding which of the significant ECHAM6 regime frequency changes are considered for the decompositions in Sect. 4.4. "

In addition, we slightly reformulated the beginning of Sect. 4.4 ( l. 425) in order to avoid the impression of relying too much on the comparison with ERA5.

-l.411: "Section" should be replaced by "section"

Done

-l.432: Add "," after "months"

Done

-l.605: I suggest better explaining here why extending this analysis to more models has not been done in this paper.

In l. 560 we now elaborate a bit more on why we only focused on one model:

"Furthermore, the results in Sects. 4.2–4.4 can differ for other PAMIP models, but conducting the decomposition method as applied in this study for each PAMIP model individually would be difficult: especially a comprehensive summary and interpretation of decomposition results for different models would be very challenging, in particular due to the fact that each model tends to simulate its distinct significant regime frequency changes in different months. Hence, the presented ECHAM6 analysis might be considered as a first step and adapting the employed decomposition methodology for a feasible implementation into a multimodel analysis might provide a prospect for future studies."

-Figure 3 caption: replace "dotted" by "hatched"

Done

-Figure 6 caption is still confusing as blue and red as supposed to both indicate favored occurrence of cold extremes. My understanding is that red indicates favored occurrence of cold extremes and blue a decreased frequency of cold extremes. This is better explained in Fig 9 caption and should be revised here. Also, in general for the different figures showing maps of the frequency of occurrence of cold extremes (Fig3, Fig6 etc.), the color choice is not quite intuitive and very confusing to me. One would expect that blue means colder conditions, and red means warmer conditions. Hence, I strongly suggest switching or changing the colors to make it easier to understand.

Thanks for the comment, we reversed the colorbar in all plots that refer to cold extremes  in order to make the plots more intuitive. (Figs.3a–e,4,6,9,S9,S10)

-Figure A2: the name of the experiments on the figure has not been updated.

Corrected.

-Figure A2, A4, A5: Please add in the caption the model name (ECHAM6) otherwise one could think that all the PAMIP models could have been used here.

Done